# UFMTrack, an Under-Flow Migration Tracker enabling analysis of the entire multi-step immune cell extravasation cascade across the blood-brain barrier in microfluidic devices

**Mykhailo Vladymyrov[1,2]\*, Luca Marchetti[1], Sidar Aydin[1], Sasha GN Soldati[1], Adrien Mossu[1], Arindam Pal[1], Laurent Gueissaz[1], Akitaka Ariga[3], Britta Engelhardt[1]**

[1]Theodor Kocher Institute, Bern, Switzerland; [2]Data Science Lab, Bern, Switzerland; [3]Laboratory for High Energy Physics, University of Bern, Bern, Switzerland

**\*For correspondence:** mykhailo.vladymyrov@unibe.ch

**Competing interest:** The authors declare that no competing interests exist.

**Sent for Review** 31 August 2023
**Preprint posted** 04 September 2023
**Reviewed preprint posted** 21 May 2024
**Reviewed preprint revised** 07 October 2024
**Reviewed preprint revised** 27 December 2024
**Version of Record published** 15 April 2025

## eLife Assessment

This work is **important** because it elucidates how immune cells migrate across the blood brain barrier. In the revised version of this study, the authors present a **convincing** framework to visualize, recognize and track the movement of different immune cells across primary human and mouse brain microvascular endothelial cells without the need for fluorescence-based imaging using microfluidic devices. This work will be of broad interest to the cancer biology, immunology and medical therapeutics fields.

**Abstract** The endothelial blood-brain barrier (BBB) strictly controls immune cell trafficking into the central nervous system (CNS). In neuroinflammatory diseases such as multiple sclerosis, this tight control is, however, disturbed, leading to immune cell infiltration into the CNS. The development of in vitro models of the BBB combined with microfluidic devices has advanced our understanding of the cellular and molecular mechanisms mediating the multistep T-cell extravasation across the BBB. A major bottleneck of these in vitro studies is the absence of a robust and automated pipeline suitable for analyzing and quantifying the sequential interaction steps of different immune cell subsets with the BBB under physiological flow in vitro. Here, we present the under-flow migration tracker (UFMTrack) framework for studying immune cell interactions with endothelial monolayers under physiological flow. We then showcase a pipeline built based on it to study the entire multistep extravasation cascade of immune cells across brain microvascular endothelial cells under physiological flow in vitro. UFMTrack achieves 90% track reconstruction efficiency and allows for scaling due to the reduction of the analysis cost and by eliminating experimenter bias. This allowed for an in-depth analysis of all behavioral regimes involved in the multistep immune cell extravasation cascade. The study summarizes how UFMTrack can be employed to delineate the interactions of CD4+ and CD8+ T cells with the BBB under physiological flow. We also demonstrate its applicability to the other BBB models, showcasing broader applicability of the developed framework to a range of immune cell-endothelial monolayer interaction studies. The UFMTrack framework along with the generated datasets is publicly available in the corresponding repositories.

## Introduction

Immune cells continuously travel throughout our body as a means of immune surveillance. Moving within the bloodstream allows for their fast transport to even distant sites but requires extravasation once they have reached their target organ. Immune cell extravasation across the vascular wall is a multistep process regulated by the sequential interaction of different signaling and adhesion molecules on the endothelium and the immune cells (*Nishihara et al., 2020*; *Marchetti and Engelhardt, 2020*). These molecular interactions mediate distinct sequential steps, namely tethering and rolling to reduce travel speed, shear-resistant arrest, polarization, and crawling of the immune cell on the luminal surface of the endothelium and, finally, immune cell diapedesis across the endothelial layer (*Marchetti and Engelhardt, 2020*; *Nourshargh and Alon, 2014*; *Nourshargh et al., 2010*).

The precise molecular mechanisms mediating the multistep immune cell extravasation in each organ depend not only on the immune cell subset but also on the specific characteristics of the vascular bed. For example, in the central nervous system (CNS), the endothelial blood-brain barrier (BBB) establishes a tight barrier that strictly controls the transport of molecules across the BBB, ensuring tissue homeostasis required for neuronal function (*Marchetti and Engelhardt, 2020*; *Engelhardt et al., 2017*). The BBB similarly controls immune cell trafficking into the CNS. Thus, accounting for these special barrier properties, unique characteristics of the multistep T-cell migration across the BBB have been described. For instance, T cells crawl for very long distances against the direction of blood flow on the surface of the BBB endothelium in search of permissive locations for diapedesis (*Bartholomäus et al., 2009*; *Lyck and Engelhardt, 2012*; *Castro Dias et al., 2021*). Research on T-cell interaction with the BBB has already been successfully translated into therapies in the clinic (*Steinman and Zamvil, 2005*; *Singer, 2017*).

Exploring the entire multistep extravasation of immune cells across the BBB has been significantly advanced by making use of in vitro BBB models maintaining their barrier properties and placing them into microfluidic devices (*Lyck et al., 2023*). Combined with microscopic setups that allow for in vitro live-cell imaging (*Figure 1A and B*) of the immune cell interaction with the brain endothelial monolayer under physiological flow over time, the molecular mechanisms mediating the sequential interaction of T cells during extravasation across the BBB have been delineated (*Marchetti and Engelhardt, 2020*). The live-cell imaging is largely performed in the phase-contrast imaging modality (*Figure 1C*), which does not require establishing fluorescent labels of the imaged cells and avoids potential phototoxicity of the fluorescent imaging modality. These studies have shown that upon their arrest, T cells polarize and either crawl at speeds between 3 and 10 µm/min over the brain endothelial monolayer or probe the endothelial monolayer by remaining rather stationary and sending cellular protrusions into the endothelial monolayer (*Figure 1D*; *Steiner et al., 2010*; *Rudolph et al., 2016*). Both behaviors can lead to diapedesis of the T cells across the brain endothelial monolayers. This process lasts at least 3–5 min, with some immune cells observed to protrude and retract several times prior to finalizing a prolonged diapedesis process to the abluminal side of the brain endothelial monolayer (*Castro Dias et al., 2021*; *Abadier et al., 2015*). Finally, T cells that have successfully migrated across the brain endothelial monolayer usually continue to migrate underneath the endothelial monolayer (*Shulman et al., 2009*; *Arts et al., 2021*).

The data analysis of these 'in vitro flow assays' requires time-consuming offline frame-by-frame analysis of the imaging data by individual experimenters, in which the dynamic interactions of each individual immune cell has to be followed over the entire time of the assay manually and assigned to specific categories. Such analysis is tedious, and accurately assigning the different T-cell behaviors requires experience. Thus, this manual analysis is prone to inevitable errors and subjective judgments of the various experiments. Furthermore, the time-consuming manual T-cell tracking limits the number of events that can be studied and, thus, the statistical power of the analysis.

Automation of the analysis of the recorded multistep T-cell extravasation across the BBB in the microfluidic device would thus be highly desirable. It is, however, hampered as these assays are usually performed with unlabeled cells and imaged by phase contrast (*Figure 1C*), which poses a challenge due to the similar grayscales and morphology of the immune cells interacting with the brain endothelial cells. Further challenges include rapid cell movement under flow. When perfused on top of the endothelial cells in the presence of shear flow, at first T cells instantly appear within the field of view (FoV). Later on they are often either suddenly displaced over a certain distance or completely detached and washed away (*Figure 1D*). Proper analysis of these events is mandatory for reliable

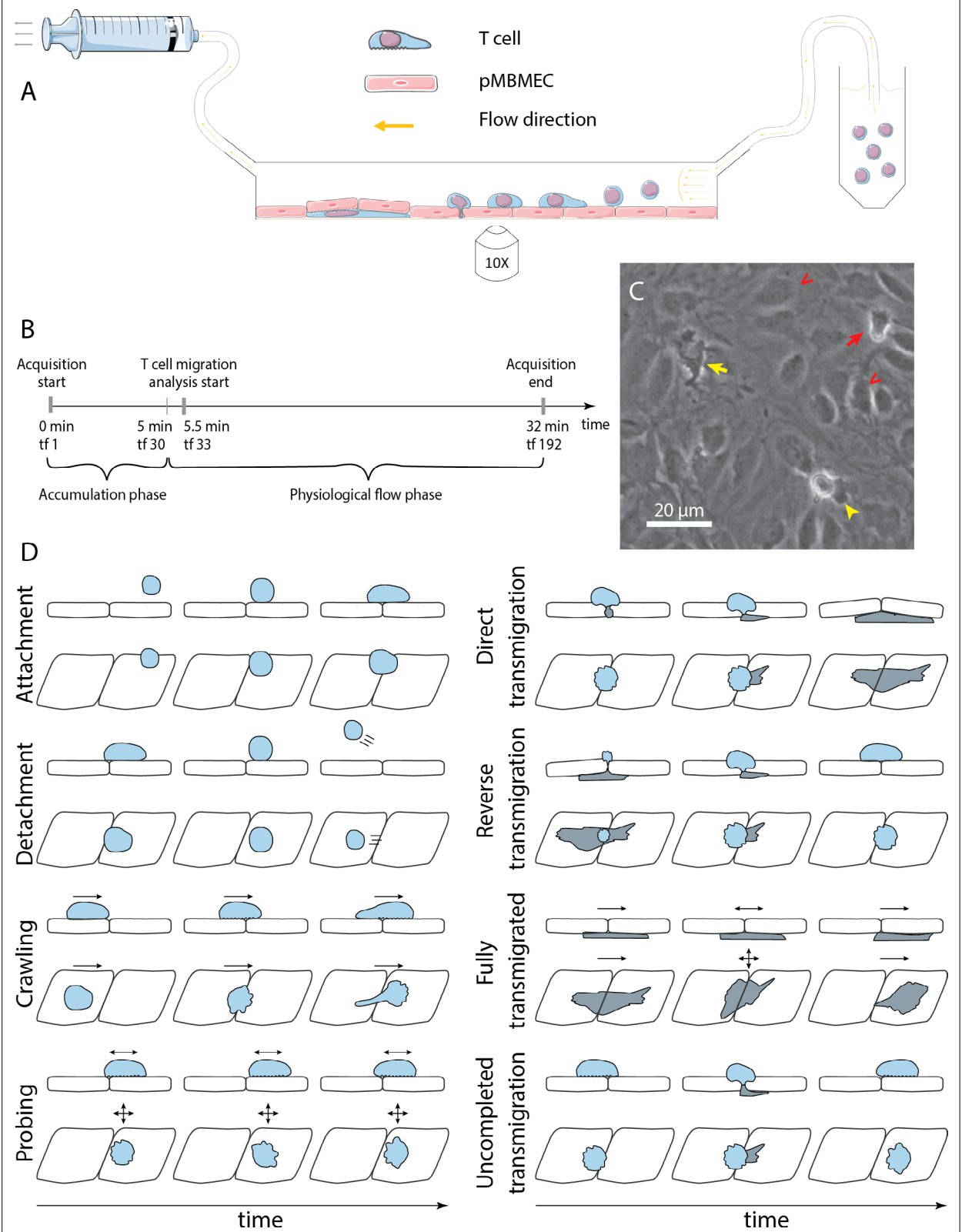

**Figure 1.** In vitro analysis of the multistep cascade of T-cell migration across the blood-brain barrier (BBB) model under physiological flow. (**A**) In vitro under-flow assay setup. T cells were perfused on top of the primary mouse brain microvascular endothelial cell (pMBMEC) monolayer, and their migration under flow was observed using phase-contrast imaging modality. Imaging was performed with a time step of 10 s/time frame. (**B**) In vitro flow assay timeline. During the accumulation phase under flow with the shear stress of 0.1 dynes/cm², T cells adhered to the pMBMEC monolayer. After 5 min

*Figure 1 continued on next page*

*Figure 1 continued*

(time frame 30), the shear stress was increased to 1.5 dynes/cm², leading to rapid detachment of not firmly adhering T cells. Analysis of the post-arrest T-cell behavior was thus starting at 5.5 min (time frame 33). tf = time frame. (**C**) Example of phase-contrast imaging data. Red arrow – crawling T cell; yellow arrow – fully transmigrated T cell; yellow arrowhead – transmigrated part of a partially transmigrated T cell; red V arrowheads – pMBMECs. Scale bar is 20 μm. (**D**) Schematic representation of distinct T-cell behavior regimes detected and analyzed using the developed under-flow migration tracker (ᵁᶠᴹTrack) framework. Crawling cells migrate continuously while probing cells interact with the pMBMECs and move around the interaction point within two cell size (20 μm) as indicated by the arrows. Side and top views are shown.

T-cell tracking but also with respect to the study of the overall avidity of the dynamic T-cell interaction steps with the underlying brain endothelium. Thus, it was compulsory to establish a tracking solution that accounts for the effect of the flow on the migrating cells and the distinct migration regimes.

Segmenting of cells imaged in phase contrast migrating on top of cellular monolayer of similar appearance is a challenging task. Furthermore, the analysis of T-cell interactions requires detection and quantification of the transmigration. While recent advances in computer vision established several frameworks for detection (*He et al., 2017*; *Wang et al., 2022*; *Carion et al., 2020*; *Schmidt et al., 2018*) or segmentation (*Ronneberger et al., 2015*; *Stringer et al., 2021*), no tool is readily available to complete both tasks simultaneously.

There is a large number of powerful cell tracking solutions developed that focus on different aspects of cell migration and are embedded in different frameworks or solutions. The general-purpose Trackpy (*Allan et al., 2021*) allows for simple particle tracking and is not suitable to track cells migrating under flow. It is used for the tracking of cells imaged in phase contrast in the Usiigaci framework (*Tsai et al., 2019*). The widely used TrackMate (*Ershov et al., 2022*) plugin for ImageJ (*Rueden et al., 2017*) allows for tracking of particles performing either the Brownian motion or a linear motion, but not the combination of two. Instant appearance or disappearance as well as the resolving tracks of under-segmented cells cannot be easily established in this framework. Similarly, CellTraxx (*Holme et al., 2023*) performs matching of cells in the adjacent frames, prohibiting its use for tracking of under-segmented, intersecting cells or under flow. The Bayesian Tracker (btrack) framework (*Ulicna et al., 2021*) is using spatial information as well as appearance information for track linking focusing on the cell divisions required for cell lineage tracing.

Here, we introduce the developed under-flow migration tracker (ᵁᶠᴹTrack) framework that systematically addresses the hurdles mentioned above, allowing it to perform automated tracking and analysis of cell-cell interactions imaged in phase contrast. We also show a successful implementation of ᵁᶠᴹTrack to build an analysis pipeline for T-cell interactions with brain microvascular endothelial cells in vitro under physiological flow. ᵁᶠᴹTrack reaches 90% T-cell tracking efficiency, performing comparably to manual analysis while eliminating the experimenter's bias and improving the accuracy of T-cell tracking. Therefore, it enables significant savings in the labor force and time for data analysis.

## Results

To design and develop the automated T-cell under-flow migration analysis framework (ᵁᶠᴹTrack framework) presented here, we made use of in vitro imaging datasets following T-cell migration across primary mouse brain microvascular endothelial cells (pMBMECs) under physiological flow in vitro. The framework combines three components: T-cell segmentation and transmigration detection, T-cell tracking under flow, and analysis of each of the multistep T-cell migration cascade steps. Segmentation and transmigration detection of the T cells, migrating on the pMBMECs, is performed with a 2D+T U-Net-like convolutional neural network (*Ronneberger et al., 2015*). The T-cell under-flow tracking algorithm was formulated as a constrained optimization problem.

Next, we describe the methods and algorithms employed to develop ᵁᶠᴹTrack. Links to the code and the datasets used for model training can be found in the Data availability section.

### T-cell segmentation

Reliable cell segmentation is crucial for reliable cell tracking. In the phase-contrast imaging modality, it was impossible to achieve reliable differentiation between T cells and endothelial cells of the pMBMEC monolayer based on pixel intensity. For detection of T-cell transmigration across pMBMEC monolayers (diapedesis), sufficiently reliable cell segmentation of the transmigrated T cells was also

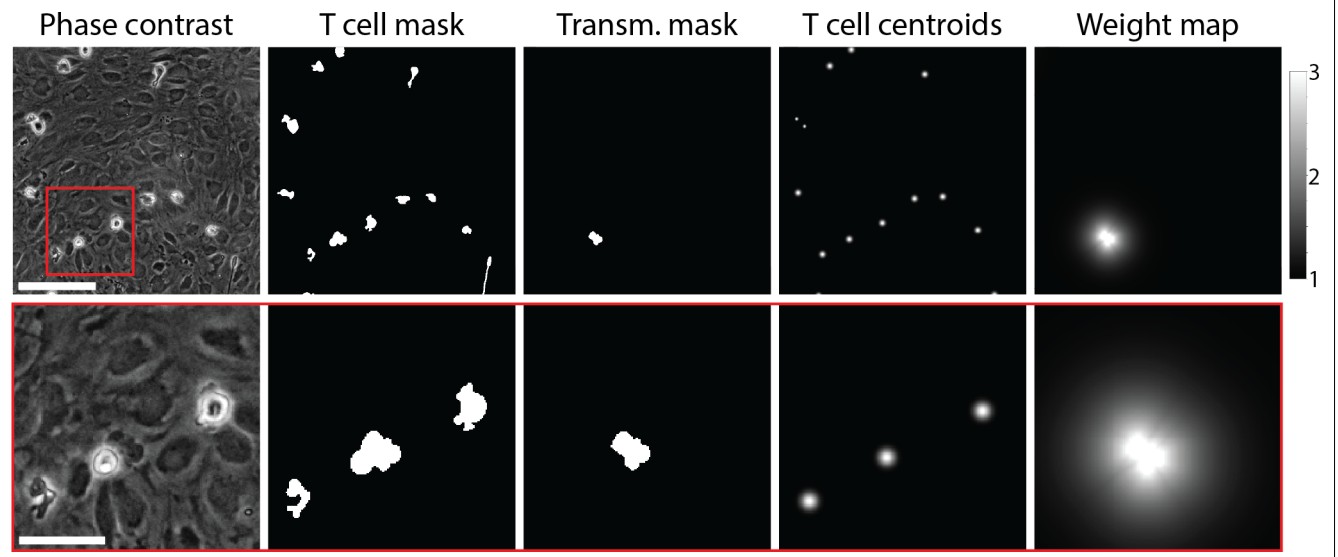

**Figure 2.** Example of data used for training of the T-cell segmentation model. From left to right: Phase-contrast microscopy image of T cells migrating on top of the primary mouse brain microvascular endothelial cell (pMBMEC) monolayer; annotated T-cell mask; annotated mask of transmigrated part of T cells; T-cell centroids; $w_{cell}$ weight map. The bottom row shows zoom-in on the highlighted area. Scale bars are 50 μm in the top row and 20 μm in the bottom row.

required. To achieve this, we have designed 2D and 2D+T U-Net-like (*Ronneberger et al., 2015*) fully convolutional neural network-based models for multitask learning. The models were trained to predict three maps: cell probability, the probability that the T cell is below the pMBMEC monolayer, and cell centroids. The models performed predictions based on the grayscale of the respective phase-contrast images in the case of the 2D model or sequences of 5 time frames in the case of the 2D+T model. The models were implemented in TensorFlow (*Abadi et al., 2016*). The training was performed on a dataset corresponding to 154 Megapixels of raw image data using the annotation mask for T cells ('T-cell mask') and the mask of the transmigrated part of the T cells ('transmigration mask'), the centroids map, and the weight map (*Figure 2*). For validation, 37 Megapixels of raw imaging data were used. The models have approximately 3 and 8 M parameters correspondingly, and the model architectures are summarized in *Appendix 1—table 1* and *Appendix 1—table 2*. Details on models' implementation, training, and image processing can be found in the 'Segmentation models' section in Appendix 1.

In the T-cell mask prediction task, the 2D+T model outperformed the 2D model by a notable 9% according to the average precision (AP) metric (*Table 1*). At the same time, for the transmigration mask, which is much more difficult to infer, the 2D model performance reached only 54%, which is insufficient for reliable detection of T cells migrating across the pMBMEC monolayer. In this task, the 2D+T model outperformed the 2D model by 32% AP. We observed that the 2D+T model was sensitive to frame misalignment, leading to false positive detection of transmigrated T cells. Thus, to generate the preliminary T-cell masks used for frame alignment and histogram normalization, we employed the 2D model. Afterward, for the T-cell segmentation and transmigration detection, we

**Table 1.** Comparison of the performance of the 2D and 2D+T models for T-cell segmentation.

| | Performance, AP | |
| --- | --- | --- |
| | 2D | 2D+T |
| T-cell mask | 86.12% | 95.26% |
| Transmigration mask | 50.56% | 82.74% |

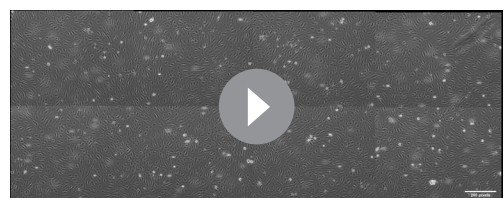

**Video 1.** Phase-contrast time-lapse image sequence of CD4 T cells interacting with IL-1β-stimulated endothelium. Eight tiles of the imaging are aligned and stitched together.

https://elifesciences.org/articles/91150/figures#video1

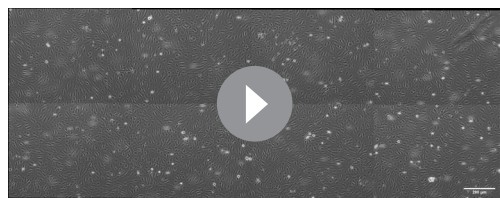

**Video 2.** Segmented T cells in the phase-contrast time-lapse image sequence of CD4 T cells interacting with IL-1β-stimulated endothelium. The mask of the segmented T cell is overlayed in red. The transmigration probability map is overlayed in yellow.
https://elifesciences.org/articles/91150/figures#video2

employed the 2D+T model, followed by a watershed algorithm based on the inferred T-cell mask and the cell centroids as seed points. A stitched phase-contrast image sequence can be seen in *Video 1* and overlayed with the segmented cells and highlighted transmigration mask in *Video 2*.

After segmentation, we suppressed the noise by discarding small objects with an area below 30 pixels (the mean area of a T cell is about 400 pixels). For each T cell, we evaluated its coordinates as geometric mean coordinates, the angle of the longer axis, and the elongation $\epsilon = \sqrt{\frac{\sum_i \left(x_i' - x_0'\right)^2}{\sum_i \left(y_i - y_0'\right)^2}}$, where $x_i', y_i'$ are projections of the $i$th T-cell mask pixel coordinates on the orthogonal long and short axes of the T cell, and $x_0', y_0'$ are the corresponding projections of the geometrical center of the T cell. The summation is performed over the area covered by the mask of the corresponding T cell. Additionally, we evaluated the mean T-cell probability $\overline{p_{cell}} = \frac{1}{n_{pix}} \sum_i p_{cell,i}$, mean transmigration probability $\overline{p_{tr}} = \frac{1}{n_{pix}} \sum_i p_{tr,i}$, and cell transmigration coefficient as $t_c = \frac{1}{n_{pix}} \sum_i \frac{min\left(p_{tr,i}, p_{cell,i}\right)}{p_{cell,i}}$ , where $n_{pix}$ is number of cell pixels and $p_{cell,i}$, $p_{tr,i}$ are the T-cell and transmigration probabilities for $i$th pixel in the predicted T-cell mask.

## T-cell tracking

The presence of flow causes several types of discontinuous events that were managed with a specialized tracking algorithm. These are: (1) the sudden appearance of T cells in the FoV, (2) the sudden detachment of a T cell followed by its disappearance from FoV, and (3) the displacement over significant distances (up to hundreds of micrometers) of T cells that do not adhere firmly to the pMBMECs. The primary inspiration for our approach was the conservation tracking algorithm (*Schiegg et al., 2013*). We could consistently reconstruct the T-cell tracks by performing global optimization constrained to controlled probabilities of T-cell appearance, disappearance, and displacement due to the flow at every timepoint. The standard tracking algorithms like tracking by association is either unable to detect track links with high displacement and thus reconstructs several disconnected segments of a cell track, or leads to a high number of false track links. Our procedure favors the reconstruction of long T-cell tracks while at the same time allowing for tracking of the T cells which detach or are displaced by the flow ('accelerated T-cell movement') over a longer distance (8 μm) with speed significantly higher than their crawling speed.

The tracking consists of four main steps: linking, searching for track candidates, global track consistency resolving, and resolving the track intersections. The inputs for the tracking are the segmented cells and the cell centroids (*Figure 3A*). The centroids and the cell proximity are used to quantify under-segmentation by introducing the node multiplicity (*Figure 3B*, see section T-cell tracking in Appendix 1 for details). At the linking step, we identified all possible connections of T cells between the time frames (*Figure 3C*).

Next, during the search for track candidates, we aimed to find continuous track segments of T cells or under-segmented groups of T cells crawling on top of or below the pMBMEC monolayer without accelerated movement segments on the T-cell track characterized by rapid T-cell displacements. The whole dataset of T cells across all timepoints was represented as a graph. Each vertex corresponds to a T cell (multiplicity m=1) or a group of potentially under-segmented T cells (m>1). The vertices are connected according to links obtained at the linking step. Track segments were found by performing global optimization to find consistent connectivity of vertices across the timepoints (*Figure 3D*) by employing an approach inspired by the conservation tracking algorithm summarized in *Schiegg et al., 2013*. Optimization was performed using the CP-SAT constrained optimization procedure using the open-source OR-Tools library (*Perron and Furnon, 2021*).

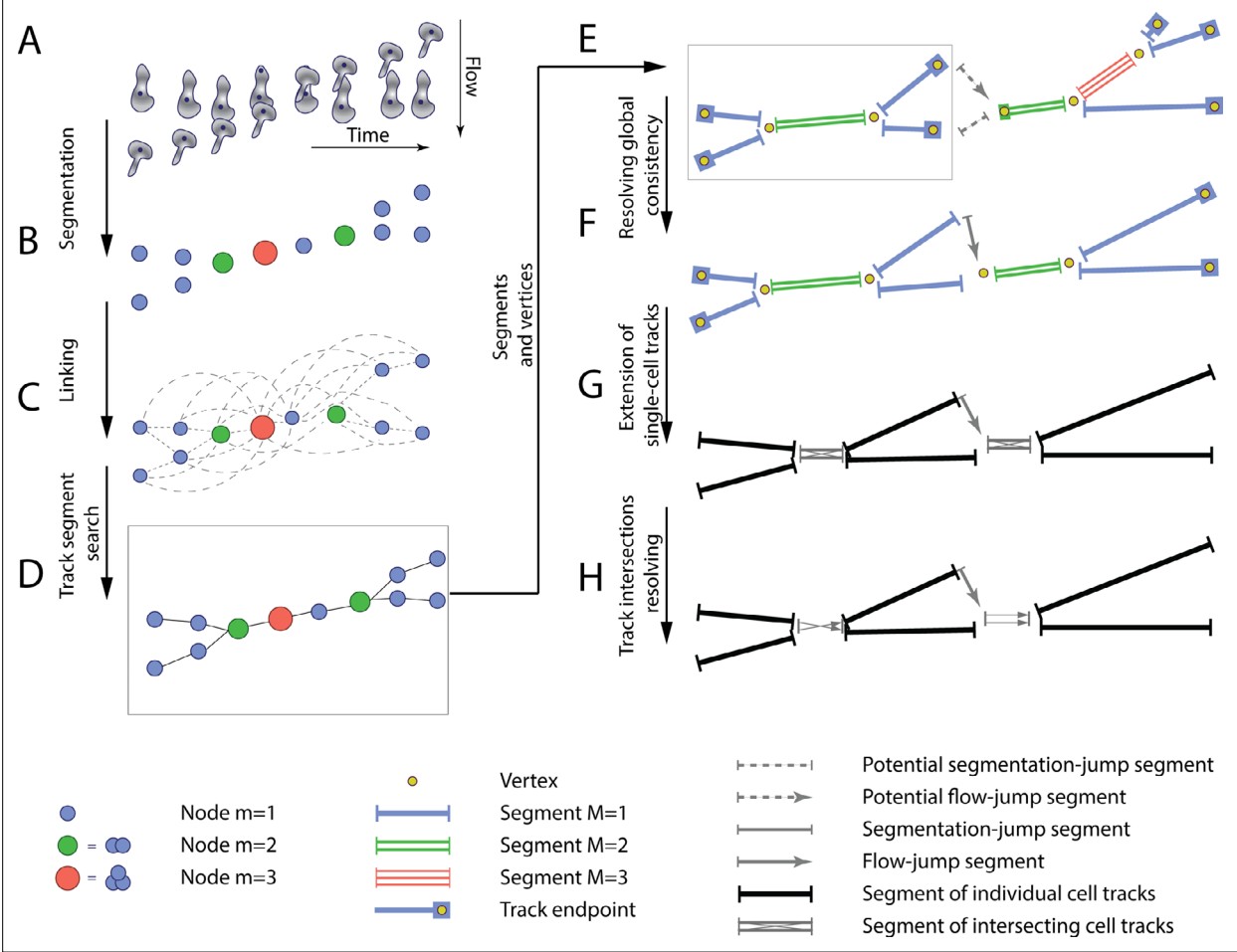

**Figure 3.** The T-cell tracking pipeline. (**A**) Segmentation and centroids of two migrating T cells under flow over time. (**B**) Nodes corresponding to segmented T cells. Node sizes and colors denote the multiplicity estimate of a node. (**C**) Nodes, together with links between adjacent in space and time nodes, form a graph. (**D**) Selected links are obtained with global optimization on the graph. (**E**) Graph of segments and vertices obtained according to node multiplicity. (**F**) Resolving global segment multiplicity consistency with global optimization on the graph. Additionally, a search for rapid displacement of segmented T cells due to under-segmentation or flow is performed. (**G**) Extension of tracks of individual T cells into the track intersection. (**H**) Resolving track intersections.

The online version of this article includes the following figure supplement(s) for figure 3:

**Figure supplement 1.** Transmigration detection.

Next, resolving global track consistency was performed. In this step, the scope of T-cell tracking is shifted from individual nodes (representing T cells at particular time frames as well as groups of under-segmented T cells) to the track segments (unambiguous sequences of nodes) and vertices at the endpoints of the segments. These are track start and end points, points of merging and separation of track segments in case of under-segmentation, as well as ambiguous points on a track. The latter was identified by sudden T-cell displacement, a hallmark of detaching and reattaching T cells and T-cells transitioning from properly segmented to under-segmented T cells or vice versa (*Figure 3E*). This was followed by the search for potential missing track segments due to accelerated T-cell movement under flow and missing links in the track crossing points. Both are characterized by significant displacement lengths such that they were not detected during the linking step. Thus, we will refer to both as 'jumps' in this section. We have also eliminated short and therefore unreliable segments, as well as segments whose multiplicity was found to be $M = 0$ (*Figure 3F*). Afterward, we separated the segments into two categories, namely segments with multiplicity $M = 1$, i.e., tracks of isolated T cells, and segments with multiplicity $M > 1$, i.e., tracks of under-segmented groups of T cells, where tracks of several T cells intersected. Finally, we extended the tracks of isolated T cells into the intersection

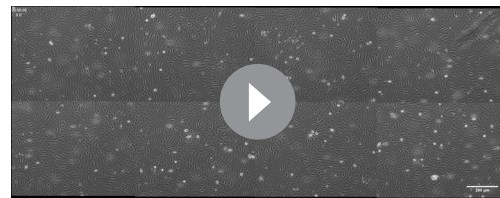

**Video 3.** Tracks of T cells reconstructed in the phase-contrast time-lapse image sequence of CD4 T cells interacting with IL-1β-stimulated endothelium. Tracks are shown after the flow was increased to a shear stress level of 1.5 dynes/cm². All videos are shown accelerated by a factor of 96 as can be seen on the timestamp label. Only tracks included in the analysis are shown (see text for details). Scale bar = 200 μm.
https://elifesciences.org/articles/91150/figures#video3

if the branching vertex had a multiplicity of $n$ and was splitting in exactly $n$ segments with multiplicity $M = 1$ (**Figure 3G**).

Lastly, we resolved the track intersections to obtain reliable T-cell tracks, i.e., identifying track segments corresponding to the same T cell before and after the under-segmented track region (**Figure 3H**).

Detailed information on the developed under-flow T-cell tracking algorithm is given in the T-cell tracking section in Appendix 1.

## T-cell migration analysis

Next, we performed the T-cell migration analysis based on the reconstructed T-cell tracks. We selected tracks inside the fiducial area of the FoV, namely coordinates of the T cell at all timepoints along the track were located at least 25 µm away from the bounding box enclosing all segmented T cells. Next, tracks of T cells touching another T cell at the end of the assay acquisition were excluded since T cells directly adjacent to each other can hide the start of T-cell transmigration across the pMBMEC monolayer and thus compromise correct detection and quantification of this step. Additionally, only tracks with T cells were assigned in at least 6 time frames during the physiological flow phase. We also require T cells to be assigned for at least 75% of time frames along the track. Under-segmented parts of T-cell tracks were not considered. Examples of selected tracks can be seen in **Video 3**.

After exclusion of the tracks of the cellular debris, which were misclassified as T cells during the segmentation step, using a dedicated 'not-a-T-cell' classifier, we performed identification of transmigration (**Figure 3—figure supplement 1**), probing and crawling migration regimes, as well as accelerated movement along the T-cell track. Detailed information on the analysis of the T-cell tracks is given in the 'T-cell migration analysis' section in Appendix 1.

Finally, for each track, we evaluated motility parameters for each of the following migration regimes: probing before the transmigration, crawling before the transmigration, all crawling above pMBMECs monolayer including T-cell crawling segments after the first transmigration attempts, all crawling below pMBMECs monolayer, whole T-cell track excluding accelerated movement and tracking inefficiency regions, as well as whole T-cell track. Specifically, we evaluated the following T-cell motility parameters: duration of each migration regime, the total vector and absolute displacements, the migration path length, the average migration speed (displacement over time), average crawling speed (path length over time), and finally the mean and standard deviation of the instantaneous speed. We evaluated migration time, displacement, and average speed for the accelerated movement regime.

## Analysis of trafficking datasets

Having developed a full pipeline based on the ᵁᶠᴹTrack framework for automated analysis of the multistep T-cell migration cascade across the BBB model, we next aimed to compare the results of the automated analysis with previous studies, assess the capacity of the framework to gain novel insight into T-cell migration under flow, and evaluate its performance as compared to manual analysis.

### CD4⁺ T-cell analysis

To this end, we first analyzed a total of 18 imaging datasets each corresponding to 32 min recording dedicated to understanding the multistep migration of CD4⁺ T cells across non-stimulated (n=6), TNF-stimulated (n=5), and IL1-β-stimulated (n=7) pMBMEC monolayers under physiological flow with the developed pipeline.

In these in vitro live-cell imaging datasets, phase-contrast imaging was performed as described in the 'Data acquisition' subsection in the Materials and methods section. The T-cell accumulation phase corresponding to the shear stress of 0.1 dynes/cm² lasted for 32 time frames (5 min), followed by conditions of physiological flow (shear stress 1.5 dynes/cm²) for the subsequent 160 time frames

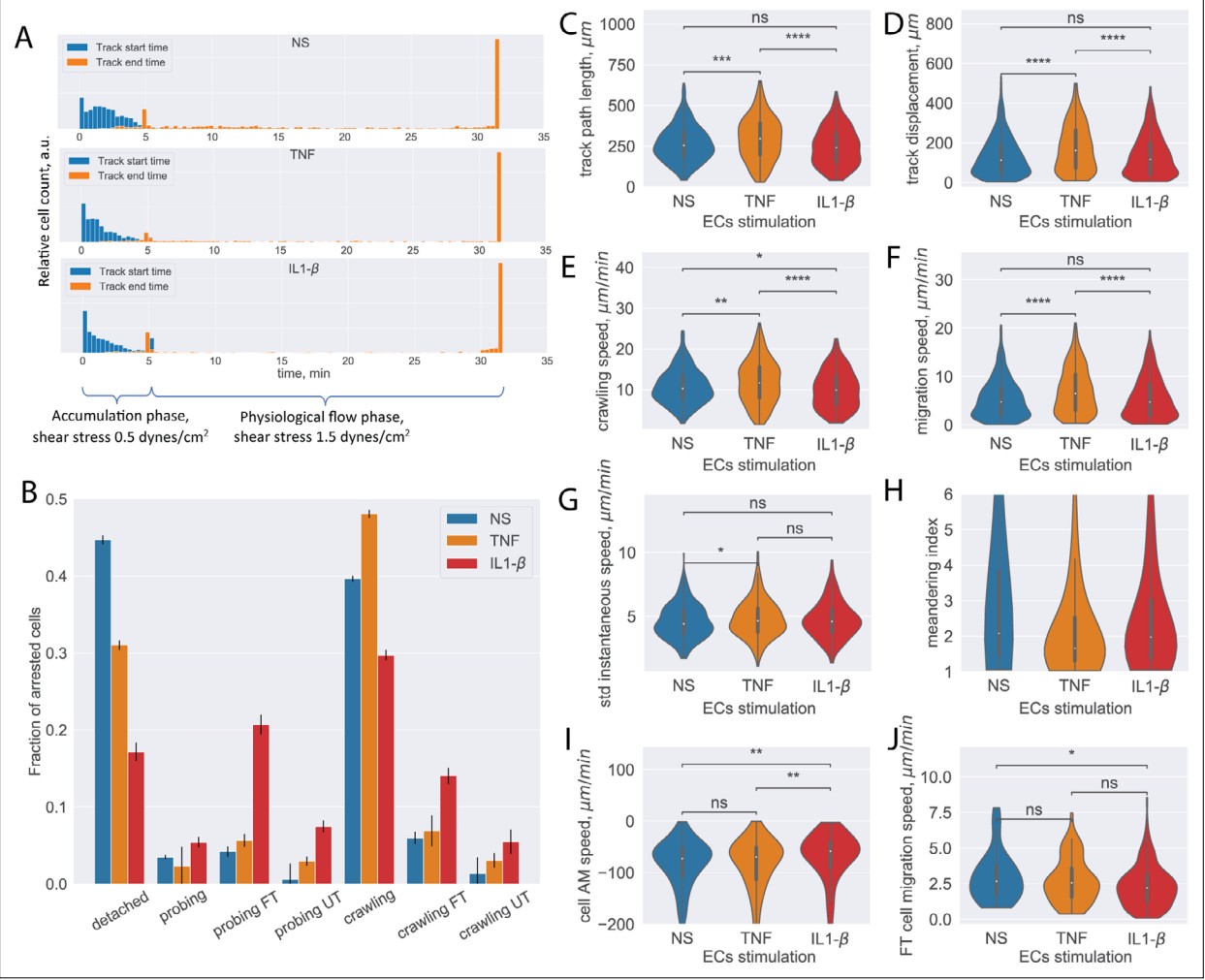

**Figure 4.** Analysis of CD4+ T-cell tracks. (**A**) Start and end time distribution of CD4+ T-cell tracks on non-stimulated or TNF- or IL-1β-stimulated primary mouse brain microvascular endothelial cells (pMBMECs). Under-flow migration tracker (UFMTrack) correctly captures increased T-cell detachment from the non-stimulated pMBMECs after 5 min when the flow is increased to physiological levels. (**B**) Quantification of CD4+ T-cell behavior in the respective categories obtained on non-stimulated and TNF- and IL-1β-stimulated pMBMECs. Error bars show the statistical error of the mean (see text for details). (**C–H**) Motility parameters of the crawling CD4+ T cells were obtained for the three endothelial stimulatory conditions. Distributions of T-cell path length (**C**), displacement (**D**), crawling speed (path/time) (**E**), migration speed (displacement/time) (**F**), variability of instantaneous T-cell crawling speed along the track (standard deviation, **G**), and meandering index (**H**). (**I**) Distribution of CD4+ T-cell accelerated movement (AM) speed is a proxy metric for the T-cell adhesion to the healthy or inflamed endothelium. (**J**) Migration speed distribution of the transmigrated CD4+ T cells. Stimulation applied to the luminal side of pMBMECs affects T-cell migration at the abluminal side of pMBMECs after their transmigration. FT – T cells performed full transmigration; UT – T cells performed uncompleted transmigration. Statistical tests are performed with the Mann-Whitney U test. Statistical significance is indicated as follows: ns – Not significant (p > 0.05); * – p ≤ 0.05; ** – p ≤ 0.01; *** – p ≤ 0.001; **** – p ≤ 0.0001.

(27 min). The shear stress is controlled by an automated syringe pump. After increasing to physiological flow rates (*Steiner et al., 2010*), a significant number of T cells detached from the pMBMEC monolayers, with higher numbers of T cells detaching from non-stimulated pMBMEC monolayers when compared to those stimulated with pro-inflammatory cytokines as detected by the distribution of the track ending times between 5 and 30 min (*Figure 4A*).

Analyzing these datasets with the established UFMTrack framework, the T-cell behavior type for each detected T-cell track and the aggregated T-cell behavior statistic were obtained for the stimulated and non-stimulated pMBMEC monolayers (*Figure 4B*). The total number of analyzed T-cell tracks is n=971, n=780, and n=1218 for the non-stimulated, TNF-stimulated, and IL1-β-stimulated pMBMEC monolayers correspondingly. The data obtained with UFMTrack were in accordance with our previous observations obtained by manual frame-by-frame analysis (*Steiner et al., 2010*; *Abadier*

*et al., 2015*; *Haghayegh Jahromi et al., 2019*). We further obtained the data for T-cell migration speed, T-cell displacement, and T-cell path lengths for the crawling T cells, as these are the primary cell motility parameters. As shown in *Figure 4C–H*, we observed statistically significant differences in the mean T-cell motility parameters depending on the pMBMECs stimulation condition. Furthermore, we observed differences in the shape of the distribution of T-cell crawling speeds, consistent with our previous reports of underlying differences in the mechanisms mediating T-cell crawling on non-stimulated vs cytokine-stimulated pMBMECs (*Haghayegh Jahromi et al., 2019*). While not statistically significant for the IL1-b and with a low significance for the TNF stimulations, an increased variance of the T-cell instantaneous speed on stimulated compared to non-stimulated pMBMECs is observed (*Figure 4G*). This suggests that T-cell crawling is often interrupted by T-cell recognizing specific cues on the stimulated endothelium. The T-cell meandering index distribution was high on non-stimulated and lower on stimulated pMBMECs (*Figure 4H*), underscoring that cytokine stimulation enhances directed T-cell movement on the pMBMEC monolayer.

Since the analysis was performed automatically with $^{UFM}$Track, it allowed for deeper insights into the T-cell migration behavior on the pMBMEC monolayers than obtained by manual frame-by-frame analysis. Specifically, the detection of accelerated movement by $^{UFM}$Track enables the researcher to quantify the kinetics of individual T-cell detachment from the endothelium rather than simply quantifying a bulk T-cell detachment rate. This detachment kinetics is reflected in the average speed experienced during accelerated movement, which is significantly lower for the IL1-b condition (*Figure 4I*), suggesting the need to break more bonds with the endothelium when pro-inflammatory cytokines are present.

As our $^{UFM}$Track workflow achieves sufficient segmentation efficiency to detect T cells below the pMBMEC monolayer and T-cell transmigration across the pMBMEC monolayer, it also enables investigation of T-cell movement after the transmigration step. In *Figure 4J*, we show the distribution of the migration speed for the transmigrated CD4$^+$ T cells. Interestingly, cytokine stimulation of pMBMECs, although applied from the luminal side, also affected T-cell movement at the abluminal side of the pMBMEC monolayer. While, in the microfluidic device used in the present study, the migration of T cells below the pMBMEC monolayer may not be biologically relevant, this analysis option will be valuable for future studies involving multilayer in vitro BBB models that include the vascular basement membrane in addition to pericytes and astrocytes to mimic the entire neurovascular unit (*Castro Dias et al., 2021*; *Mossu et al., 2019*).

Importantly, our $^{UFM}$Track automatically detects and characterizes unusual events, such as reverse T-cell transmigration, that are easily overlooked with manual counting. We do not present the statistics here, as only a few such events were observed. However, when applied to the multilayer BBBs in forthcoming in vitro models, systematic detection and analysis of these rare events will be important to understand T-cell migration in immune surveillance.

## CD8$^+$ T-cell analysis

As we have previously shown that the multistep T-cell extravasation across pMBMEC monolayers differs between CD4$^+$ and CD8$^+$ T cells (*Rudolph et al., 2016*), we next analyzed two datasets studying the multistep migration of CD8$^+$ T cells across non-stimulated and TNF/interferon-gamma (TNF/IFN-γ)-stimulated pMBMEC monolayers under physiological flow over 161 time frames (27 min). The CD8$^+$ T cells were slightly different in size and appearance when compared to the CD4$^+$ T cells used for training the segmentation model. To benchmark our established $^{UFM}$Track pipeline in this more difficult configuration, the datasets were first manually analyzed by four experimenters: one advanced experimenter with 4 years of experience (AdEx) and three inexperienced experimenters who received comparable 2 hr introduction and training (Ex1-Ex3). The analyses were performed on the subset of the acquisition, 161 time frames long starting from time frame 31.

### CD8$^+$ T-cell analysis

Manual cell analysis and tracking were performed as described in the 'Materials and methods' section separately for each of the 8 tiles of the tiled acquisition. Next, all crawling CD8$^+$ T cells that did not perform transmigration were manually tracked for the time span after the flow was increased to the physiological level using the manual tracking in ImageJ.

First, we compared the CD8[+] T-cell behavior statistics obtained manually and by our automated analysis pipeline (*Figure 5A–C*). The results obtained by the automated analysis were in full agreement with the manual analysis performed by the experienced experimenter. At the same time, the data highlight the variability of the results from the inexperienced experimenters, confirming the potential for significant experimenter bias.

We also show the detection efficiency of T cells by the [UFM]Track. To achieve this goal, we combined the numbers of CD8[+] T cells detected in each tile by manual analysis. Due to the overlap between the individual tiles, some CD8[+] T cells are seen more than once. To obtain an estimate of the total CD8[+] T-cell count detected manually, we scaled the number of CD8[+] T cells accordingly to the number of detected unique T-cell tracks (see next paragraph). By this approach, we found T-cell detection efficiency to be above 90% (*Figure 5D*). The increased T-cell density can explain lower efficiency in the non-stimulated condition. These findings set the appropriate T-cell density for automated analysis of their migration behavior on pMBMEC monolayers to 200–300 cells per FoV of the size of 3.8 mm$^2$ or 50–80 cells/mm$^2$.

Next, we compared the T-cells tracks as obtained manually by three experimenters as well as automatically by our [UFM]Track. To this end, we first matched the tracks in adjacent tiles to obtain the tracks on the whole imaging area, avoiding multiple counts of the same track. This was achieved by pattern matching with initial offsets between tiles obtained by an automatic frame alignment procedure performed as part of the automatic analysis. We considered T-cell tracks observed in different tiles to be tracks of the same T cell if $f_{in} > \frac{1}{2}f_{out} - 0.8$, where $f_{in}$ is the fraction of timepoints along a track at which the distance between the T cells was below 17 µm and $f_{out}$ was the fraction of timepoints along a track at which distance between cells was above 25 µm. We used the same approach to match the T-cell tracks obtained by each experimenter and the automated [UFM]Track. To compare performance in an objective manner, we excluded manually obtained tracks that lay outside of the fiducial volume of the automated analysis, as well as T cells that were touching another T cell at the end of the acquisition, as those were also excluded from the automated analysis. We then took as 100% the sum of all T-cell tracks detected manually. We evaluated the fraction of all T-cell tracks detected by each experimenter as well as automatically by our [UFM]Track.

In *Figure 5E*, we show that our pipeline achieves a comparable T-cell tracking efficiency when compared to the manual analysis. While its performance was not superior to that of the experimenter with 4 years of expertise in the analysis of the under-flow datasets, it does perform better than less experienced experimenters.

We also compared the T-cell motility parameters obtained manually and automatically for the non-stimulated and stimulated pMBMECs (*Figure 5F–I*). Data obtained for the CD8[+] T-cell migration speed on pMBMECs was comparable between all experimenters and the automated analysis. In contrast, results obtained for the T-cell crawling speed were significantly different between the automated approach and manual analysis. Taking a closer look at the T-cell tracks (*Figure 5—figure supplement 1*), we readily observed that the manual analysis contains significant errors in the assignment of the T-cell position. This leads to a jittery pattern in the T-cell migration tracks and overestimates the T-cell path and, thus, T-cell crawling speed. Employing our novel automated analysis pipeline eliminated this systematic error in the measurements.

To finally investigate the potential benefit in the time required for automated vs manual data analysis, we examined the time required for a given experimenter to analyze such datasets manually using the Clockify time tracker (*Clockify, 2022*). In *Figure 5J*, we show the average time investment needed to perform a complete T-cell behavior analysis and T-cell tracking in each imaging tile, as well as on average. Clearly, the experienced experimenter outperforms the inexperienced experimenters by a factor of 3. On average, a researcher would thus spend 8.1 hr analyzing a dataset with 300T cells when only the tracks of crawling T cells (50%, i.e. 150 cells) are analyzed. If all T-cell tracks were to be analyzed, this time would further increase to 12.9 hr. Given that on average 10 datasets can be produced per day, manual analysis becomes a bottleneck, leading to delays in exhaustive data analysis and thus ultimately in research progress. The [UFM]Track framework reduces analysis time by a factor of 3 when analyzing only crawling cells and 5 if all cell tracks are to be analyzed (*Figure 5K*). With an average analysis time of 2.3 hr, the 10 experiments carried out in 1 day can be thoroughly analyzed within 1 day of machine time. This enables the scalability of flow-based immune cell migration experiments while simultaneously lifting the burden of tedious and time-consuming manual analysis from the researchers.

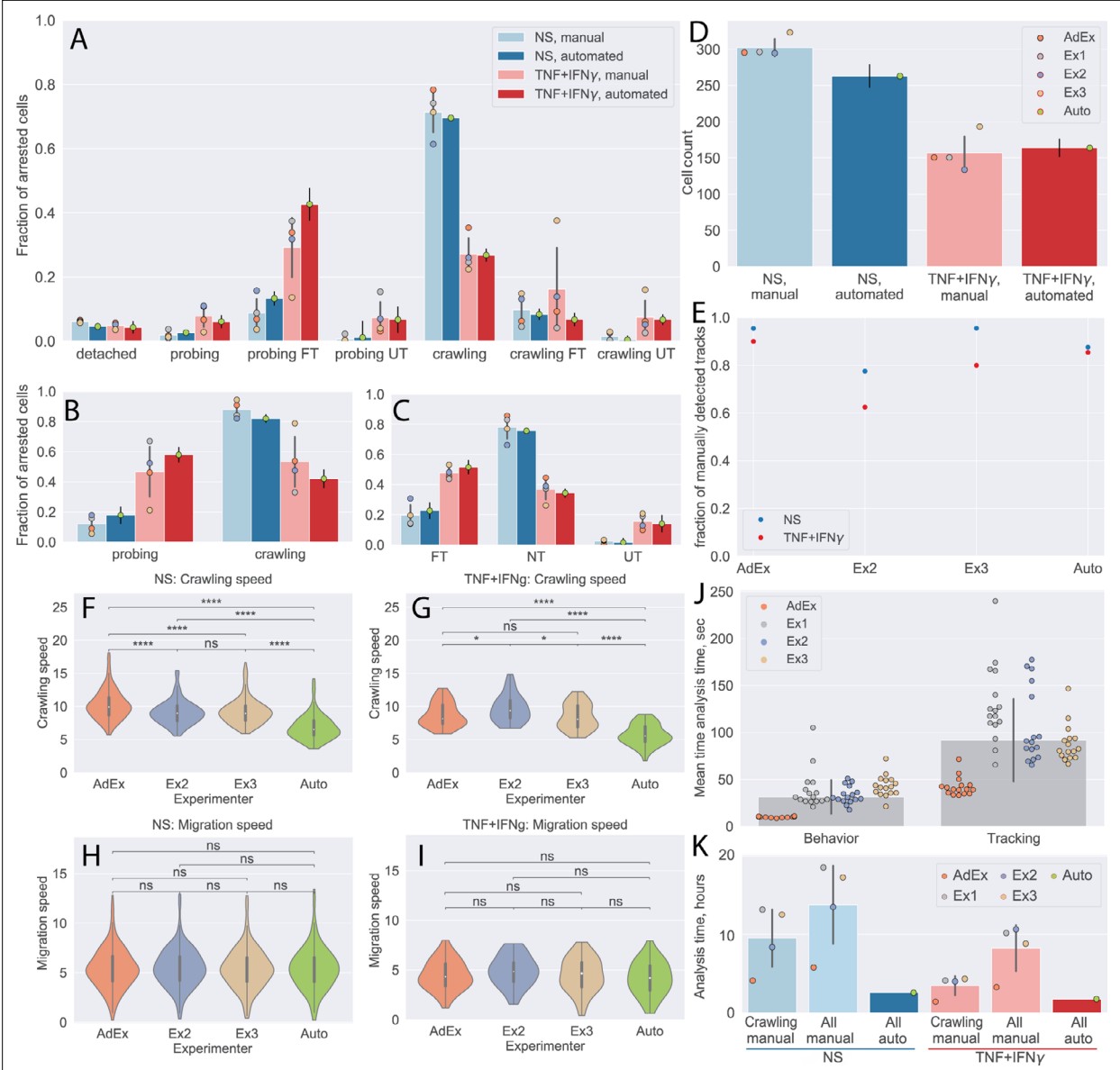

**Figure 5.** Comparison of automated analysis with under-flow migration tracker (UFMTrack) and manual analysis of the CD8+ T-cell tracks. (**A–C**) CD8+ T-cell behavior statistic obtained for non-stimulated (NS) and cytokine-stimulated primary mouse brain microvascular endothelial cells (pMBMECs) as obtained manually by four experimenters, as well as automatically with UFMTrack. (**A**) Quantification of CD8+ T-cell behavior in the respective categories obtained on non-stimulated and TNF/interferon-gamma (TNF/IFN-γ)-stimulated pMBMECs is consistent with results obtained by manual frame-by-frame analysis. Cytokine stimulation of pMBMECs increases T-cell probing behavior (**B**) and T-cell transmigration rate (**C**). Error bars show the standard deviation of the manual analysis and the statistical error of the mean for automated analysis. Points correspond to individual experimenters. (**D**) Counts of CD8+ T cells were obtained manually by four experimenters and automatically by UFMTrack. The T-cell detection efficiency is above 90%. Error bars show the standard deviation of the manual analysis and the statistical error of the mean for the automated analysis. Points correspond to individual experimenters. (**E**) Detection efficiency of crawling CD8+ T-cell tracks between the manual and automated analysis. (**F–I**) Comparison of CD8+ T-cell crawling speed (path/time) (**F, G**) and migration speed (displacement/time) (**H, I**) on non-stimulated (**F, H**) and cytokine-stimulated (**G, I**) pMBMECs. The T-cell position assignment error in manual tracking leads to biased crawling speed estimation. (**J**) Comparison of the analysis time (per cell) required for behavior analysis and tracking of CD8+ T cells. (**K**) Total analysis time (per dataset) for behavior analysis and tracking of CD8+ T cells. Comparison is shown for manual analysis with tracking of crawling cells only (crawling manual), the time estimate for manual analysis with tracking of all cells (all manual), and the in-depth automated analysis of all cell tracks with UFMTrack (all auto). FT – T cells performed full transmigration; UT – T cells performed uncompleted transmigration; NT – T cells did not perform transmigration. Statistical tests are performed with the Mann-Whitney U test. Statistical significance is indicated as follows: ns – Not significant (p > 0.05); * – p ≤ 0.05; ** – p ≤ 0.01; *** – p ≤ 0.001; **** – p ≤ 0.0001.

The online version of this article includes the following figure supplement(s) for figure 5:

**Figure supplement 1.** Comparison of reconstructed T-cell tracks with the result of manual analysis.

# Applicability of <sup>UFM</sup>Track to different models of immune cell-endothelial cell interactions

Next we have assessed the broader applicability of the <sup>UFM</sup>Track framework to different cell typesand BBB models often employed for the in vitro under-flow cell trafficking assays. To this end, we have performed the automated analysis of the multistep T-cell migration cascade in a set of datasets (one to eight) across different cell types and assays. Specifically, we attempted to verify the feasibility of analysis of datasets where the cells forming the monolayer or the migrating cells have noticeably different appearances, cells from other species are used, specifically human-derived cells, and datasets with a different imaging modality. These include human peripheral blood mononuclear cells (hPBMCs), human T cells, bone marrow-derived macrophages (BMDM) interacting with a monolayer of mouse- and human-derived brain microvascular endothelial cells as well as with immobilized recombinant BBB adhesion molecules (see 'Materials and methods' section for details). To assess the performance of the model in each condition, we have performed a visual inspection of the cell masks, transmigration masks, and cell tracks for few datasets in each assay type. We then analyzed cell behavior distribution or motility parameters relevant in each case.

## hPBMC on HBMEC monolayer

The interaction of hPBMCs with the BBB is altered under different disease conditions. We have therefore analyzed a dataset of human PBMCs on human brain microvascular endothelial cells (HBMEC) to quantify the cell behavior statistic. While the endothelial cells look significantly different than the pMBMECs, the segmentation and transmigration detection quality was sufficient to perform the cell migration analysis. This was also confirmed by the visual inspection of the cell and transmigration masks and cell tracking results (*Figure 6A*). We then quantified the PBMC behavior distribution and the crawling and migration speed of the crawling PBMC (*Figure 6B and C*).

## BMDM on the pMBMEC monolayer

Invasion of macrophages into the CNS contributes to many neurological disorders. Therefore, we have explored if <sup>UFM</sup>Track also allows us to analyze the interaction of BMDMs with the BBB under physiological flow in vitro. To this end, we have analyzed two datasets of mouse BMDMs interacting with pMBMEC monolayers (*Figure 6D*). Despite the different behavior and appearance of the macrophages as compared to the T cells, the segmentation quality was sufficient to perform the cell tracking and analyze cell motility parameters (*Figure 6E and F*), as confirmed by visual inspection of the macrophages masks and cell tracking results (*Figure 6D*). Nonetheless, due to the different appearance of the flattening crawling cells and the transmigrated cells, the transmigration detection trained on the mouse T cells does not perform sufficiently well for the robust detection of transmigrating macrophages.

## Human T cells on EECM-BMEC monolayer

Next, we analyzed eight datasets of human T cells interacting with the human induced pluripotent stem cell (hiPSC) derived extended endothelial cell culture method brain microvascular endothelial cell (EECM-BMEC)-like cell monolayers (*Figure 6G*). EECM-BMECs establish barrier properties comparable to primary human BMECs and display a mature immune phenotype allowing to perform studies for the first time in a fully autologous manner of immune cell interactions with patient-derived in vitro models of the BBB (*Nishihara et al., 2021*). Despite the cells being of different species, the T cells were successfully segmented, and their transmigration was detected, as confirmed by visual inspection of the T-cell and transmigration masks and cell tracking results (*Figure 6G*).

We have confirmed the differences in migration statistics (*Figure 6J and K*), migration and crawling speed of the crawling T cells (*Figure 6H*), as well as the previously reported difference in relative (normalized to track path length) displacement (*Figure 6I*) of the T cells treated with anti-α4-integrin+anti-β2-integrin blocking antibodies (bab) and control (ctrl) condition (*Soldati et al., 2023*).

## Human T cells on immobilized recombinant BBB adhesion molecules

Finally, we have analyzed datasets acquired in a different imaging modality. Seven datasets of human T cells interacting with immobilized recombinant BBB adhesion molecules, namely ICAM-1, VCAM-1, and their combination (*Figure 6I*). In this case, the fluorescently labeled cells were imaged with

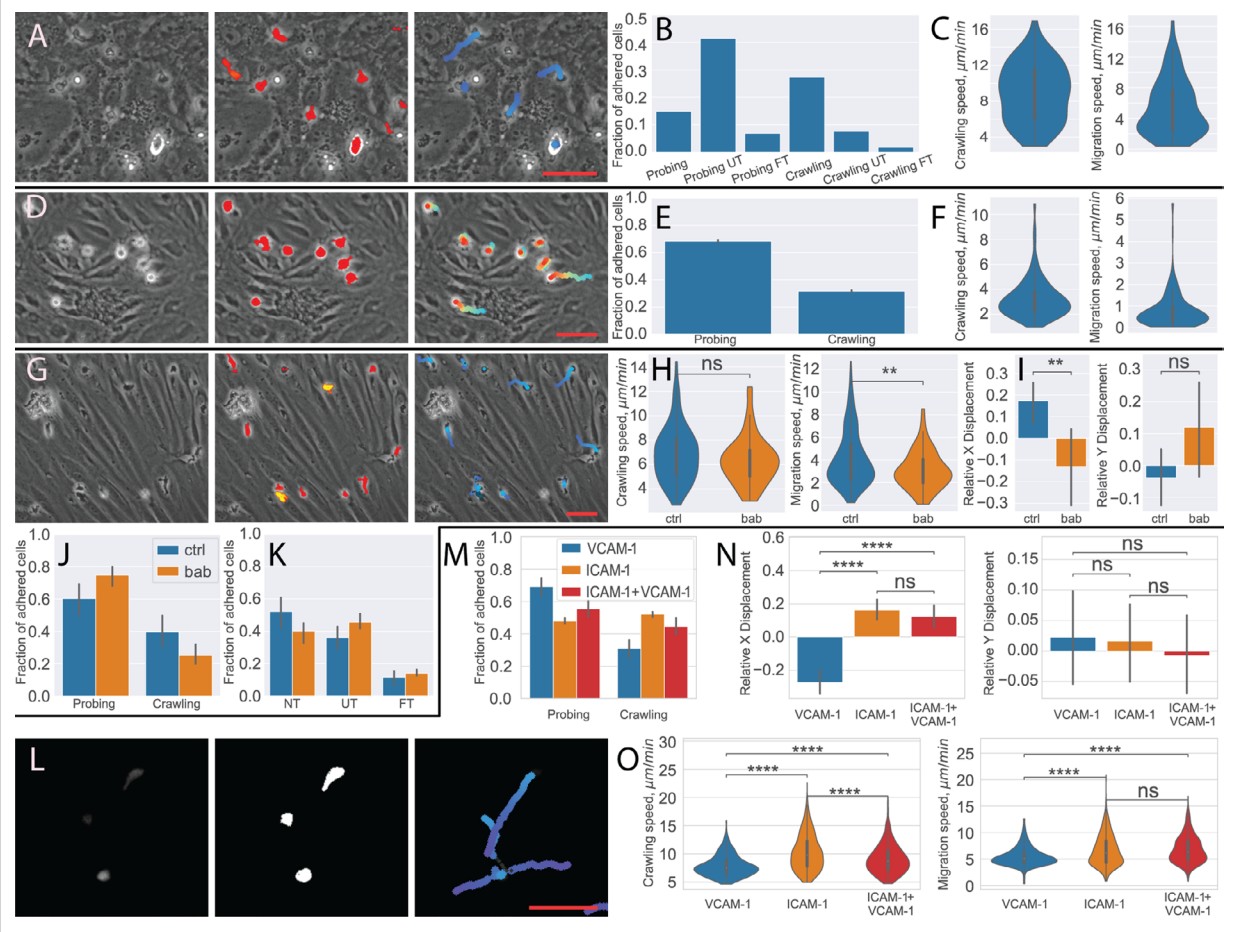

**Figure 6.** Analysis of different blood-brain barrier (BBB) models with under-flow migration tracker (^UFM^Track). (**A–C**) Human peripheral blood mononuclear cells (PBMCs) interacting with human brain microvascular endothelial cells (HBMEC). (**A**) Raw image frame, segmentation, and tracking results. (**B**) PBMCs behavior distribution. (**C**) Crawling and migration speed distribution. (**D–F**) Bone marrow-derived macrophages (BMDM) interacting with the primary mouse brain microvascular endothelial cell (pMBMEC) monolayer. (**D**) Raw image frame, segmentation, and tracking results. (**E**) BMDM crawling vs probing behavior distribution. (**F**) Crawling and migration speed distribution. (**G–K**) Human T cells interacting with the extended endothelial cell culture method brain microvascular endothelial cell (EECM-BMEC) monolayer. (**G**) Raw image frame, segmentation, and tracking results. (**H**) Crawling and migration speed distributions for the control condition (ctrl) and with the blocking anti-bodies (bab). (**I**) X and Y relative cell displacement. (**J**) Crawling vs probing behavior comparison. (**K**) Transmigrations behavior comparison. (**L–O**) Human T cells interacting with immobilized recombinant BBB adhesion molecules. (**L**) Raw fluorescent image frame, thresholding-based segmentation, and tracking results. (**M**) Crawling vs probing behavior distribution. (**N**) X and Y relative displacement (**O**). Crawling vs probing speed comparison. Scale bars = 50 µm. Statistical tests are performed with the Mann-Whitney U test. Statistical significance is indicated as follows: ns – Not significant (p > 0.05); * – p ≤ 0.05; ** – p ≤ 0.01; *** – p ≤ 0.001; **** – p ≤ 0.0001.

The online version of this article includes the following figure supplement(s) for figure 6:

**Figure supplement 1.** Comparison of sizes of crawling cells.

epi-fluorescent microscopy. In the absence of a monolayer, transmigration detection is not necessary. Thus, we performed cell segmentation directly on the fluorescent signal (see Materials and methods for details) using the same segmentation strategy as for the T-cell masks. The tracking was then performed the same way as for other experiments. The T cells were successfully segmented and tracked, as confirmed by visual inspection. We compared T-cell behavior and motility parameters for three conditions of the recombinant adhesion molecules: ICAM-1, VCAM-1, and ICAM1+VCAM1 (*Figure 6M and O*). Consistent with previous observations (*Steiner et al., 2010*; *Soldati et al., 2023*) where the analysis was performed using the ImageJ (*Rueden et al., 2017*) either manually or with the TrackMate (*Ershov et al., 2022*) plugin, we observed that in the absence of ICAM-1, T cells were moving in the direction of flow (*Figure 6N*). At the same time, ^UFM^Track allows for capturing the rapid accelerated motions of the cells and thus reconstructing longer continuous T cell tracks. This enables

capturing the migrations properties of the consecutive steps of the T-cell migration cascade on the cell-by-cell basis.

## Discussion

In this study, we have developed the $^{UFM}$Track framework. It consists of independent modules and allows for segmentation (*Figure 2*), tracking (*Figure 3*), and motility analysis of immune cells, migrating on, across, and below (*Figure 1D*) the monolayer of brain microvascular endothelial cells under physiological flow in vitro. The developed method relies exclusively on phase-contrast imaging data (*Figure 1B*). Therefore, it does not require establishing fluorescent labels of the migrating immune cell population to be studied, avoids potential phototoxicity to be considered for fluorescent imaging modalities (*Jae Oh et al., 1999*; *Saetzler et al., 1997*; *Purschke et al., 2010*), and is thus the preferred choice for analyzing the trafficking of sensitive cell types. T-cell segmentation and prediction of the transmigrated T-cell areas are performed using a custom 2D+T U-Net-like convolutional neural network. It enables reliable segmentation of T cells both above and below the pMBMEC monolayer. The existing particle and cell tracking toolkits consider migration of one cell type and thus are not suitable for the detection of distinct migration regimes and cell interactions (*Allan et al., 2021*; *Tsai et al., 2019*; *Ershov et al., 2022*; *Ulicna et al., 2021*). Furthermore, to our knowledge, none of the existing algorithms consider migration under the flow causing rapid cell displacement. The tracking of T-cell interactions with the pMBMECs during all migration regimes under physiological flow required designing a new tracking algorithm considering rapid T-cell appearance, disappearance, and displacement in the FoV caused by the flow. We have also developed approaches that resolve track intersections, i.e., identifying track segments corresponding to the same T cell before and after under-segmented track regions, which are inevitable during T-cell migration. By establishing the detection of T-cell crawling, probing, transmigration, and accelerated movement combined with reliable T-cell tracking, we have enabled the in-depth analysis of distinct migration regimes on a cell-by-cell basis. By reducing the dataset analysis time by a factor of 5, $^{UFM}$Track allows for performing a thorough analysis of 10 experiments that a researcher can carry out in 1 day within 1 day of machine time. We have demonstrated that the automated analysis performs on par with manual analysis while improving accuracy and eliminating experimenter bias, and enables scalability of flow-based immune cell migration experiments by reducing the analysis cost and lifting the burden of time-consuming manual analysis.

In this work, we have demonstrated the applicability of the developed framework to the analysis of the multistep extravasation of CD4[+] and CD8[+] T cells across non-stimulated or cytokine-stimulated pMBMEC monolayers under physiological flow. We have also demonstrated that the developed framework allows for automated analysis of T-cell behavior statistics and motility parameters of distinct T-cell migration regimes. Results of the automated analysis of CD4[+] T-cell behavioral statistics performed with $^{UFM}$Track are in agreement with previous studies using manual data analysis (*Figure 4B*; *Abadier et al., 2015*). The automated analysis of datasets of CD8[+] T-cell migration has shown comparable performance with manual analysis performed by one experimenter with 4 years of analysis experience and three less-experienced experimenters. At the same time, the variance of the results obtained manually showed significant experimenter bias that the automated analysis eliminated (*Figure 5F*). Additionally, the automated analysis allowed for in-depth analysis of all T-cell migration categories and precise evaluation of T-cell motility parameters, which was not achieved by manual T-cell tracking, even by the most experienced user.

We have further demonstrated the applicability of the $^{UFM}$Track framework to the analysis of different cell types (BMDM, PBMC), species (human-derived T cells, HBMEC, and EECM-BMEC), as well as tracking and analysis of cell migration imaged using the fluorescent imaging modality. In all cases, the cells could be successfully segmented and tracked, and the cell migration analysis was performed on a cell-by-cell basis. The main challenge for the system is reliable segmentation of the T cells and the transmigrated T cells. In the case of HBMEC the monolayer has a dotted pattern unlike the rather smooth pMBMEC. This leads to more false positives in the T-cell segmentation at the level of individual frames, which nonetheless could be efficiently filtered at the level of cell tracking. In the case of the BMDMs, since they have significantly different appearances during transmigration from the T cells, reliable transmigration detection was not obtained. This prevents the system from quantifying the transmigration statistics of the BMDMs, yet the cell detection and thus motility parameter evaluation of BMDMs was successful (*Figure 6D–F*). For all the other analyzed cell types, both the

cell segmentation and transmigration detection model trained on the annotated CD4[+] T cells also produced transmigration maps of sufficient quality for reliable transmigration detection. The framework was able to successfully detect the cells of sizes that differ by almost a factor of two (*Figure 6—figure supplement 1*). This demonstrates that, simply by discriminating cells by size, it is possible to extend [UFM]Track to study the interaction of several types of immune cells migrating on top of a cellular monolayer under flow and imaged by phase contrast.

Quantifying the fraction of transmigrated T cells is crucial for studying the molecular mechanisms governing the infiltration of autoaggressive T cells across the BBB into the CNS parenchyma and the immune surveillance. The role of specific endothelial or T-cell adhesion molecules in influencing the dynamic interactions of T cells with pMBMECs can, e.g., be probed by quantifying the ratio of T-cell crawling behavior to T-cell probing (*Figure 6M*; *Abadier et al., 2015*; *Wimmer et al., 2019*; *Aydin et al., 2023*). The analysis procedure established in the [UFM]Track framework is flexible and can be easily extended to evaluate further properties of the migrating cells. For example, the number of times a T cell interrupts its crawling regime by switching to short probing behavior on the endothelium can be evaluated, thus providing information on the distribution of 'hot spots' on the endothelium. Additionally, experiment scalability and the analysis on a cell-by-cell basis enable the search for distinct populations of T cells, e.g., according to the distribution of probing to crawling behavioral ratios. The strength of T-cell adhesion to the endothelium can be probed with the developed framework using the measure of T-cell detachment rate and the distribution of the previously overlooked T-cell accelerated movement occurrences and accelerated movement speed. Finally, the possibility of quantifying the motility parameters of the transmigrated T cells enables future studies involving multilayer in vitro BBB models that include the vascular basement membrane in addition to mural cells such as pericytes and astrocytes mimicking the entire neurovascular unit.

While in this work we have focused primarily on the analysis of mouse T cells interacting with pMBMEC under flow or other immune cells interacting with the endothelial monolayer, the methods we present establish a foundation for a broad range of studies involving in vitro under-flow studies of immune cell trafficking. The tracking algorithm and motility analysis developed in the [UFM]Track framework are also directly applicable to the analysis of immune cell interactions with recombinant BBB adhesion molecules under flow. In this case, either phase-contrast or epi-fluorescent imaging can be employed when studying fluorescently labeled immune cells, as demonstrated. The T-cell segmentation based on deep neural networks can be applied to studies of the trafficking of other immune cell subsets on and across the pMBMEC monolayer. We demonstrated the successful application of the [UFM]Track to cells with significantly different appearances, such as the HBMEC cellular monolayers.

We make [UFM]Track available to the community to build upon it for addressing their specific research questions. The modularity of the framework simplifies the adaptation to analysis of different cell types. The application to immune cell trafficking across other endothelial monolayers, including those from different species or vascular beds as well as lymphatic endothelial cells where the cell appearance is drastically different from the ones demonstrated here, is possible but requires fine-tuning of the trained segmentation model. While training the models presented here required a large dataset of annotated T-cell masks, and transmigration masks, this can be largely avoided for future development. Future models can be developed more efficiently by leveraging our existing annotated dataset of T-cell migration on the pMBMECs while employing transfer learning approaches in a multitask framework with weak supervision and fluorescent labels as auxiliary learning targets to adapt the model for the segmentation of cells with different appearances. This can be further improved by adopting self-supervised contrastive or masked learning methods, which have demonstrated significant advancements in model pretraining in recent years (*Caron et al., 2021*; *Archit et al., 2023*; *Kraus et al., 2024*; *Xie et al., 2023*). This is the subject of our future studies. The framework could be employed in combination of other methods, e.g., by incorporating fluorescently labeled endothelial adherens junctions one could differentiate the trans- from the paracellular transmigration in the migrating cells. In this case the transmigration locations detected by the [UFM]Track framework could be automatically extracted for analysis. One could employ the tracking under flow implemented in the [UFM]Track framework for tracking the immune cells migrating on the vascular wall in vivo to capture the accelerated movement of cells. By sharing the data and open-source code of the [UFM]Track framework, i.e., the training data used for the segmentation model training, trained models, as well as the model architecture, and full

under-flow T-cell tracking and migration analysis pipeline, we hope to encourage the community to pursue these developments to advance the field.

One current limitation of the method is the performance reduction of the T-cell tracking when the density of migrating T cells significantly increases (>250 cells/dataset). Thus, the recommended T-cell concentration is about 100–150 k/ml. However, to avoid nonphysiological interactions between migrating T cells and T-cell clumping, the T-cell density should be kept at moderate levels anyway. Additionally, analyzing data from 8 stitched FoVs allows for obtaining comparable statistics. The current analysis of the behavioral statistics quantifies the fraction of different T-cell migration regimes with respect to the number of adhered cells instead of the total T-cell count passing through the microfluidic device. While the number of fast-moving T cells during the accumulation phase cannot be directly counted using the imaging modality employed here, it can be estimated indirectly from the imaging data with additional calibration experiments and a dedicated machine learning model. Another limitation is that currently, we do not consider dividing immune cells. While cell division events happen rarely (<1% of T cells) and are not primary events for the study of immune cell interaction with the BBB model, the modular architecture of our framework will facilitate future extension to detect cell division.

By enabling experiment scalability, unbiased analysis with advanced accuracy, and an in-depth analysis of large datasets of T-cell dynamics under flow, the computational and analytical framework presented here contributes to the 3R principle when studying the interaction of cells derived from animal models by reducing the number of animals to be sacrificed. $^{UFM}$Track can be employed for fundamental research of the molecular mechanisms governing immune cell trafficking across a range of vascular beds, screening of pharmaceutical treatments, as well as for personalized medicine based on the evaluation of treatment efficacy on patient-derived T-cell migration behavior on patient-derived endothelial monolayers. One can furthermore envisage that $^{UFM}$Track can be suitable to study cancer cell metastasis across different endothelial cell monolayers. Eventually, the developed framework can be extended to real-time operation during image acquisition. Combined with transgenic photo-convertible immune cells allowing for the photoconversion of immune cells according to their behavior will allow for subsequent fluorescent cell sorting and scRNA-Seq analysis. Such advanced studies are needed to reveal the role of genetic differences governing T-cell migration regimes.

## Materials and methods

### pMBMEC culture

pMBMECs were isolated from 8- to 12-week-old C57BL/6J WT mice and cultured exactly as described before (*Steiner et al., 2010*; *Coisne et al., 2013*). Intact monolayers were stimulated or not with 10 ng/mL of recombinant mouse TNF, 20 ng/mL of recombinant mouse IL1-β, or 5 ng/mL recombinant mouse TNF+100 U/mL recombinant mouse IFN-γ 16–24 hr prior to the assays as previously described (*Haghayegh Jahromi et al., 2019*).

### Human brain microvascular endothelial cells

HBMEC were kindly provided by Prof. Nicholas Schwab (University of Münster, Germany). They were cultured to confluency on speed coating solution coated µ-Dishes (35 mm, low, iBidi) for 2 days as previously described (*Haghayegh Jahromi et al., 2019*).

### EECM-BMEC-like cells

The Extended Endothelial Cell Culture Method (EECM) was used to differentiate human induced pluripotent stem cells (hiPSCs) to brain microvascular endothelial cell (BMEC)-like cells, what were then employed as a human in vitro model of the BBB. In brief, hiPSCs from one healthy control (cell line ID: LNISi002-B) were established from erythroblasts in the laboratory of Renaud DuPasquier (University of Lausanne, Switzerland). EECM-BMEC-like cells were cultured to confluency on collagen IV and fibronectin-coated µ-Dishes (35 mm, low, iBidi) for 2 days (*Soldati et al., 2023*).

### T-cell and hPBMC preparation

Naïve CD4$^+$ and CD8$^+$ T-cell isolation: Peripheral lymph nodes and spleens from 2D2 and OT-I C57BL/6J mice were harvested, and single-cell suspensions were obtained by homogenization and

filtration through a sterile 100 μm nylon mesh. A second filtration was applied after erythrocyte lysis (0.83% $NH_4Cl$, Tris-HCl). 2D2 and OT-I cells were isolated respectively with magnetic $CD4^+$ and $CD8^+$ T-cell selection beads (EasySep, STEMCELL Technologies).

In vitro activation of naïve $CD8^+$ T cells: OT-I $CD8^+$ T cells were activated as described before (*Rudolph et al., 2016*; *Aydin et al., 2023*). Activated $CD8^+$ T cells were cultured in IL-2-containing media for 3 days post-activation.

In vitro activation of naïve $CD4^+$ T cells: 2D2 $CD4^+$ T cells were activated as described before (*Haghayegh Jahromi et al., 2019*). Activated $CD4^+$ T cells were cultured in IL-2-containing media for 24 additional hours.

hPBMC were isolated from buffy coats of healthy donors by Ficoll-Paque Plus (Cytiva) density gradient and were frozen and stored in a liquid nitrogen tank until use as previously described in *Soldati et al., 2023*.

Human $CD4^+$ T cells were isolated by employing a $CD4^+$ T-cell isolation kit (Miltenyi Biotec kit) following the provider's instructions. Subsequently, effector/memory $CD4^+$ T cells were sorted by fluorescence-activated cell sorting into different Th subsets according to their specific surface expression of chemokine receptors ($CCR6^-CXCR3^+CCR4^-$ for Th1; $CCR6^+CXCR3^+CCR4^-$ for Th1*; $CCR6^-CXCR3^-CCR4^+$ for Th2; $CCR6^+CXCR3^-CCR4^+$ for Th17) and expanded as previously described (*Nishihara et al., 2020*; *Mossu et al., 2019*; *Wimmer et al., 2019*; *Engen et al., 2014*; *Sallusto et al., 1998*; *Zielinski et al., 2012*).

## BMDM preparation

BMDMs were prepared as described in Berve et al. (manuscript in preparation). In brief, the mouse bone marrow was harvested and washed through a 100 μm nylon mesh. After 7 days of cell culture the cells were plated at a density of $17–20×10^6$ cells/mL in BMDM medium supplemented with 5 ng/mL of recombinant mouse macrophage colony stimulating factor (mCSF, R&D Systems, Minneapolis, USA, MN, 416-ML-500) onto non-treated 100 mm tissue culture Petri dishes (Greiner Bio-One, St. Gallen, Switzerland) for 7 days at 37°C and 5% $CO_2$. The cells were harvested by incubation with 0.05% Trypsin.

## T-cell binding assays under physiological flow conditions

For live-cell imaging of T-cell interaction with recombinant BBB cell adhesion molecules under the physiological flow condition, μ-Dishes (35 mm, low, iBidi) were coated with 1.54 μg/mL human recombinant VCAM-1 (BioLegend) and/or 1.14 μg/mL human recombinant ICAM-1 (R&D Systems) in DPBS for 1 hr at 37°C and blocked with 1.5% (vol/vol) bovine serum albumin (Sigma-Aldrich) in DPBS overnight at 4°C.

CMFDA pre-labeled Th cells were incubated with the isotype control antibody (30 μg/mL) for 30 min at 37°C (8% $CO_2$) prior to imaging (exact antibody concentrations are illustrated in the figure legend). The cells were subsequently perfused on top of the pre-coated dishes at a concentration of $10^6$/mL in MAM as previously described (*Steiner et al., 2010*; *Soldati et al., 2023*).

## In vitro under-flow T-cell migration assay

We studied the multistep T-cell migration across monolayers of pMBMECs in a custom-made microfluidic device under physiological flow by in vitro live-cell imaging according to the previously established procedure (*Abadier et al., 2015*; *Haghayegh Jahromi et al., 2019*). The custom-made flow chamber was described in depth before (*Lyck et al., 2023*; *Coisne et al., 2013*). The flow chamber was placed on top of the pMBMECs, previously cultured in Ibidi μ-Dish to confluency, and connected to the flow system filled with migration assay medium (MAM) (*Steiner et al., 2010*). During the accumulation phase, T cells at a concentration between 55 and 166 k/mL were perfused on top for 5 min under low shear stress of 0.1 dynes/$cm^2$, allowing them to settle on top of the pMBMEC monolayer. We used lower T-cell concentration compared to previous studies to enable automated T-cell interaction analysis. Afterward, the flow was increased to physiological levels with a flow shear stress of 1.5 dynes/$cm^2$ to study postarrest T-cell behavior on pMBMECs for 27 min (*Figure 1A and B*).

## Data acquisition

The time-lapse imaging was performed during both accumulation and physiological flow phases using phase-contrast imaging at a frame rate of 6 or 12 frames/min, subsampled to 6 frames/min and resolution of 0.629 µm/pixel with a ×10 objective. In this modality the acquired images are gray scale, and the T cells, especially after the migration across the pMBMEC monolayer, have a similar appearance to the pMBMECs, making the T-cell segmentation task very challenging (*Figure 1C*). The data was acquired in tiles of 870×650 µm$^2$ (1389×1041 pixels) with an overlap of 100 µm, leading to a total acquired image area of 3170×1220 µm$^2$. For each experiment, the dataset consists of 30 time frames of T-cell accumulation and 162 time frames of dynamic T-cell interactions with the pMBMEC monolayer under physiological flow. The time step between sequential time frames for each tile is 10 s. The total acquired area is thus limited by the acquisition speed or by the flow chamber size.

The HBMEC dataset was acquired as previously described (*Soldati et al., 2023*). In brief, 2 tiles were acquired with a total size of 3891×1504 pixels with pixel size 0.69 µm and frame rate 6 frames/min. The accumulation phase was 4 min, and the physiological flow phase was 5 min.

The BMDM dataset was acquired similarly to the T cell interacting with the pMBMEC datasets. In brief, 1 tile was acquired with a size of 2048×1504 or 1388×1040 pixels with pixel size 0.69 or 0.629 µm and frame rate 6 frames/min. The accumulation phase lasted 6 min, and the physiological flow phase was 25 min.

The EECM-BMEC dataset was acquired as previously described (*Soldati et al., 2023*). In brief, 2 tiles were acquired with a total size of 3892×1504 or 2638×1048 pixels with pixel size 0.69 or 0.629 µm and frame rate 6 frames/min.

The dataset of human T cells on immobilized recombinant BBB adhesion molecules was acquired as previously described (*Soldati et al., 2023*). In brief, 2 tiles were acquired with a total size 3891×1504 pixels with pixel size 0.69 µm and frame rate 6 frames/min. The accumulation phase was 4 min, and the physiological flow phase was 5 min. In this dataset, there is only one compartment, thus excluding any transmigration of the T cells. We, therefore, have employed the preprocessed fluorescent images instead of the cell probability map and transmigration map produced by the segmentation neural network. In this case, the centroids are defined as local maxima localized on the preprocessed fluorescent images. The fluorescent images were preprocessed first by performing Gaussian blur filtering in 2D with a standard deviation of 0.75. Next, thresholding was applied to eliminate the background noise, clipping high brightness values (color values with abundance <0.3% of the threshold abundance), scaling to 0–255 8 bit, and saving in a lossless format. Finally, the 2 tiles were stitched together by matching the integrated color distribution in the X direction in the overlapping area.

## Manual analysis of T-cell migration

Manual cell analysis and tracking were performed according to the previously established procedure using ImageJ software (ImageJ software, National Institute of Health, Bethesda, MD, USA) (*Rudolph et al., 2016*). The number of arrested T cells was thus counted at time frame 33 of the subset. The behavior of arrested T cells was defined and expressed as fractions of arrested T cells set to 100% as follows:

- T cells that detached during the observation time ('detached').
- T cells that migrated out of the FoV detached during the observation time ('out of FoV').
- T cells that continuously crawled on the pMBMEC monolayer ('crawling').
- T cells that remained in the same location (displacement less than twice the cell size) while actively interacting with the pMBMEC monolayer ('probing').
- T cells that crossed the pMBMEC monolayer with or without prior crawling ('crawling full transmigration' and 'probing full transmigration'). The event of T-cell transmigration across the pMBMECs monolayer became obvious due to the change of appearance of the transmigrated part of the T cells from phase bright (on top of the pMBMECs monolayer) to phase dark.
- T cells that partially crossed the pMBMEC monolayer, then retracted the protrusions and continued to migrate above the monolayer ('crawling uncompleted transmigration' and 'probing uncompleted transmigration').

### <sup>UFM</sup>Track performance evaluation

The inference of T-cell masks and transmigration masks was performed on a dual-CPU Intel(R) Xeon(R) CPU E5-2670 v3 @ 2.30 GHz, 256 GB RAM node equipped with 8 Graphical Processing Units (GPU) NVIDIA GeForce GTX TITAN X/GeForce GTX 1080. Frame alignment and segmentation were performed on an Intel(R) Core(TM) CPU i7-4771 @ 3.50 GHz, 32 GB RAM workstation with NVIDIA GeForce GTX TITAN GPU. Cell tracking was performed on a dual-CPU Intel(R) Xeon(R) CPU E5-2643 v2 @ 3.50 GHz, 256 GB RAM workstation.

### Performance analysis

The statistical tests are performed with the Mann-Whitney U test. Performance of the segmentation models was evaluated based on the pixel classification metrics evaluated using the scikit-learn library (*Pedregosa et al., 2011*). Specifically, the AP is evaluated on the test set as the area under the precision-recall curve, as the weighted mean of precisions achieved at each threshold, with the change in recall from the previous threshold used as the weight. The segmentation thresholds for the cell mask and the transmigrated cell mask predictions were selected to maximize the F1 score.

## Acknowledgements

We thank Kristina Berve for providing BMDM datasets for analysis with the <sup>UFM</sup>Track. We owe sincere thanks to Dr. James McGrath and Danial Ahmad (Rochester University, NY, USA) for their detailed and insightful feedback on our manuscript. This work was supported by an UniBe ID Grant to MV, BE, and AA and the Microscopy Imaging Center (MIC) of the University of Bern.

## Additional information

### Funding

| Funder | Grant reference number | Author |
| --- | --- | --- |
| University of Bern | UniBe ID Grant | Mykhailo Vladymyrov<br>Akitaka Ariga<br>Britta Engelhardt |

The funders had no role in study design, data collection and interpretation, or the decision to submit the work for publication.

### Author contributions

Mykhailo Vladymyrov, Conceptualization, Data curation, Software, Formal analysis, Supervision, Funding acquisition, Validation, Investigation, Visualization, Methodology, Writing – original draft, Project administration, Writing – review and editing; Luca Marchetti, Data curation, Investigation, Methodology, Writing – review and editing; Sidar Aydin, Sasha GN Soldati, Data curation, Formal analysis, Methodology, Writing – review and editing; Adrien Mossu, Data curation; Arindam Pal, Laurent Gueissaz, Formal analysis; Akitaka Ariga, Conceptualization, Resources, Supervision, Funding acquisition, Methodology, Project administration; Britta Engelhardt, Conceptualization, Resources, Supervision, Funding acquisition, Writing – original draft, Project administration, Writing – review and editing

### Author ORCIDs

Mykhailo Vladymyrov ![ORCID] https://orcid.org/0000-0002-5720-9740
Sidar Aydin ![ORCID] https://orcid.org/0000-0001-7791-7931
Arindam Pal ![ORCID] https://orcid.org/0000-0002-3784-7868
Laurent Gueissaz ![ORCID] https://orcid.org/0009-0008-8695-7992
Britta Engelhardt ![ORCID] https://orcid.org/0000-0003-3059-9846

### Ethics

Animal procedures were approved by the Veterinary Office of the Canton Bern (permit no BE72/2015, BE77/2018, BE73/2021) and are in line with institutional and standard protocols for the care and use of laboratory animals in Switzerland.

Reviewer #1 (Public review): https://doi.org/10.7554/eLife.91150.4.sa1
Reviewer #2 (Public review): https://doi.org/10.7554/eLife.91150.4.sa2
Author response https://doi.org/10.7554/eLife.91150.4.sa3

# Additional files

## Supplementary files
MDAR checklist

## Data availability
The source code for the T-cell segmentation models, data preprocessing, training, performance evaluation, and inference scripts are available on GitHub (*Vladymyrov, 2025a*). This repository also contains win64 binaries used for the watershed-based segmentation of the predicted T cell probability maps and transmigration probability maps. The source code for the under-flow T-cell tracking and migration analysis, along with the scripts used for performance evaluation of the framework, is available on GitHub (*Vladymyrov, 2025b*). The training phase-contrast data with manual annotations, the reference datasets for histogram normalization, and trained models are available on Zenodo: [https://doi.org/10.5281/zenodo.7489557]. The datasets used for evaluation of our framework are available on Zenodo: [https://doi.org/10.5281/zenodo.7489972, https://doi.org/10.5281/zenodo.7489984, https://doi.org/10.5281/zenodo.7489996, https://doi.org/10.5281/zenodo.7490012, https://doi.org/10.5281/zenodo.7490020, https://doi.org/10.5281/zenodo.7490029].

The following datasets were generated:

| Author(s) | Year | Dataset title | Dataset URL | Database and Identifier |
|---|---|---|---|---|
| Vladymyrov M, Marchetti L, Aydin S, Soldati S, Mossu A, Pal A, Gueissaz L, Ariga A, Engelhardt B | 2022 | UFMTrack T-cell segmentation models and training data | http://doi.org/10.5281/zenodo.7489557 | Zenodo, 10.5281/zenodo.7489557 |
| Vladymyrov M, Marchetti L, Aydin S, Soldati S, Mossu A, Pal A, Gueissaz L, Ariga A, Engelhardt B | 2022 | UFMTrack phase contrast imaging dataset of CD4 T cell interaction with pMBMEC, part 1 | http://doi.org/10.5281/zenodo.7489972 | Zenodo, 10.5281/zenodo.7489972 |
| Vladymyrov M, Marchetti L, Aydin S, Soldati S, Mossu A, Pal A, Gueissaz L, Ariga A, Engelhardt B | 2022 | UFMTrack phase contrast imaging dataset of CD4 T cell interaction with pMBMEC, part 2 | http://doi.org/10.5281/zenodo.7489984 | Zenodo, 10.5281/zenodo.7489984 |
| Vladymyrov M, Marchetti L, Aydin S, Soldati S, Mossu A, Pal A, Gueissaz L, Ariga A, Engelhardt B | 2022 | UFMTrack phase contrast imaging dataset of CD8 T cell interaction with pMBMEC | http://doi.org/10.5281/zenodo.7489996 | Zenodo, 10.5281/zenodo.7489996 |
| Vladymyrov M, Marchetti L, Aydin S, Soldati S, Mossu A, Pal A, Gueissaz L, Ariga A, Engelhardt B | 2022 | T-cell segmentation, tracking and migration analysis of the CD4 T cells with UFMTrack, part 1 | http://doi.org/10.5281/zenodo.7490012 | Zenodo, 10.5281/zenodo.7490012 |
| Vladymyrov M, Marchetti L, Aydin S, Soldati S, Mossu A, Pal A, Gueissaz L, Ariga A, Engelhardt B | 2022 | T-cell segmentation, tracking and migration analysis of the CD4 T cells with UFMTrack, part 2 | http://doi.org/10.5281/zenodo.7490020 | Zenodo, 10.5281/zenodo.7490020 |

*Continued*

| Author(s) | Year | Dataset title | Dataset URL | Database and Identifier |
|---|---|---|---|---|
| Vladymyrov M, Marchetti L, Aydin S, Soldati S, Mossu A, Pal A, Gueissaz L, Ariga A, Engelhardt B | 2022 | T-cell segmentation, tracking and migration analysis of the CD8 T cells with UFMTrack | http://doi.org/10.5281/zenodo.7490029 | Zenodo, 10.5281/zenodo.7490029 |

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

## Appendix 1

### Segmentation models

#### Training data

Deep neural networks achieve high performance but require a large amount of data for model training. Thus, in addition to preparing a large high-quality training dataset of annotated T cells, we employed additional learning targets and data augmentation to improve model performance. To train the models, we have manually annotated subsets from four independent experiments, combined corresponding to approximately 30 min of time-lapse acquisition of 880×660 µm², 226 Megapixels in total. The data was split into nonadjacent training and validation datasets of 154 and 37 Megapixels correspondingly. We have annotated the masks of whole T cells ('T-cell mask') and the mask of the transmigrated part of the T cells ('transmigration mask'). Annotation masks were normalized to have the values 0 and 1. Additionally, the center points of the touching and overlapping T cells were annotated, while for the rest of the T cells, the center points were obtained as the center of mass of the T-cell mask. Afterward, maps of centroids were generated where each centroid was represented by higher values in the map following a two-dimensional Gaussian function with width $\sigma = \frac{1}{10} r_{cent}$, where $r_{cent}$ is the distance to the nearest centroid. This width was then clipped to the range $\sigma \in [1.5, 4]$ pixels so that all centroids have a comparable scale. To prioritize the T-cell segmentation quality in the vicinity of the transmigrated part of the T cells, we created distance maps to the nearest transmigrated T-cell pixel $r_{tm}$. We then created a weight map as $w_{cell} = 1 + \alpha_w \, exp \left( -\frac{r_{tm}}{d_{tm}} \right)$. The two annotation masks, the centroids map, and the weight map (***Figure 2***) were used in the model loss function (see Training section below).

Consecutive frames of the time-lapse phase-contrast imaging datasets were aligned by a cross-correlation and global offset optimization analogous to the one described in ***Vladymyrov et al., 2020***. Additionally, we have applied histogram normalization of the phase-contrast images (***Gonzalez and Woods, 2008***) to achieve a coherent brightness distribution across all datasets. During histogram normalization, we adjusted the brightness for all pixels such that the brightness distribution of the pMBMECs monolayer in each image matched the brightness distribution of the pMBMECs monolayer in a reference image. We have selected one dataset as a reference according to the best validation performance of trained models. Then, following common practice, we have standardized the brightness distribution according to brightness values between the 2.5th and 97.5th percentile.

During training, we performed data augmentation consisting of random rotation by arbitrary angle, brightness change (<50%), contrast change (<50%), and affine deformations. During rotation and deformation transformations, the image and all label masks were transformed coherently, employing bilinear interpolation for the input image, centroids, and the weight map. For the T-cell masks and the transmigration masks, the 'nearest' interpolation was applied. At each training iteration, a new random transformation was applied to a randomly selected crop of the training dataset, thus producing an unlimited number of possible variations.

#### Training

We trained the model in a multitask learning framework by optimizing the loss function

$$L = w_{cell} \, L_{cell} + L_{tm} + \alpha \, L_{cent},$$

where $w_{cell}$ is the weight map prioritizing T-cell segmentation quality in the vicinity of the transmigrated part of the T cells. Here,

$$L_{cell} = -\Sigma_b \left( 1_c \log \widehat{p_c} + \left( 1 - 1_c \right) \log \left( 1 - \widehat{p_c} \right) \right) / n_b$$

is the mean cross-entropy for the T-cell mask predicted probability $p_c$ given the ground truth label $l_c$;

$$L_{tm} = -\Sigma_b \left( 1_{tm} \log \widehat{p_{tm}} + \left( 1 - 1_{tm} \right) \log \left( 1 - \widehat{p_{tm}} \right) \right) / n_b$$

is the mean cross-entropy for the transmigrated T-cell mask prediction probability $p_{tm}$ given ground truth transmigration label $l_{tm}$;

$$L_{cent} = \Sigma_b \left( \left| \widehat{h_{cent}} - l_{cent} \right| \right) / n_b$$

is the $L_1$ distance between the predicted value on the centroids map $h_{cent}$ and corresponding label $l_{cent}$. The summation was performed over all $n_b$ pixels in the training batch.

Coefficients $\alpha = 5$, $\alpha_w = 2$, and the characteristic distance from the nearest transmigrated T cell $d_{tm} = 15\mu m$ were chosen empirically to maximize the intersection-over-union (IoU) metric value in the validation.

The models were trained on image crops with the size of $256 \times 256$ pixels, using Adam optimizer (*Kingma and Ba, 2014*) with a batch size of 2, for 40k iterations (~840 epochs). The training took 17 hr for 2D and 29 hr for the 2D+T models using a single GeForce GTX TITAN X GPU. The learning rate was set to $1.5 \times 10^{-3}$. During training, the learning rate was adjusted in stages. It was ramped up starting 1/4 of the nominal value for 1k iterations and then halved after 20k iterations, as shown in *Appendix 1—figure 1*. The training was stopped when the validation IoU metric reached its maximum for the T-cell mask prediction. The loss evolution along the training is shown in *Appendix 1—figure 1*.

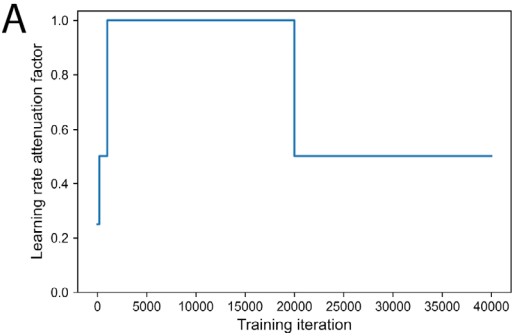
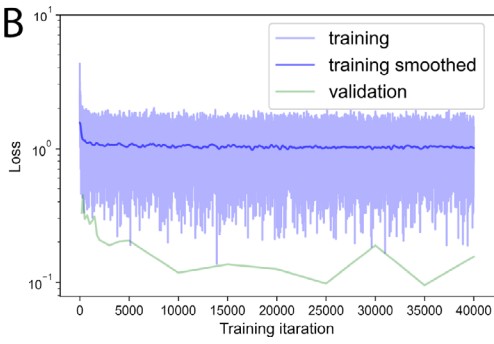

**Appendix 1—figure 1.** Training curves. (**A**) Learning rate attenuation along model training. (**B**) Loss value evolution along model training.

## Performance evaluation

Next, we evaluated the pixel-wise performance of the trained T-cell segmentation models for prediction of the T-cell mask and the transmigration mask on the validation dataset. We assessed the F1 score, Jaccard index, and AP, as shown in *Appendix 1—figure 2* and *Table 1* for the T-cell and transmigration masks for both the 2D vs 2D+T models. For the T-cell mask prediction, the 2D+T model outperformed the 2D model by a notable 9%. At the same time, for the transmigration mask, which is much more difficult to detect, the 2D model performance reached only 54%, which is insufficient for the reliable detection of T cells migrating across the pMBMEC monolayer. In this task, the 2D+T model outperformed the 2D model by 32% AP. We observed that the 2D+T model was sensitive to frame misalignment, leading to false positive detection of transmigrated T cells. Thus, to generate the preliminary T-cell masks used for frame alignment and histogram normalization, we employed the 2D model. Afterward, for the T-cell segmentation and transmigration detection, we employed the 2D+T model.

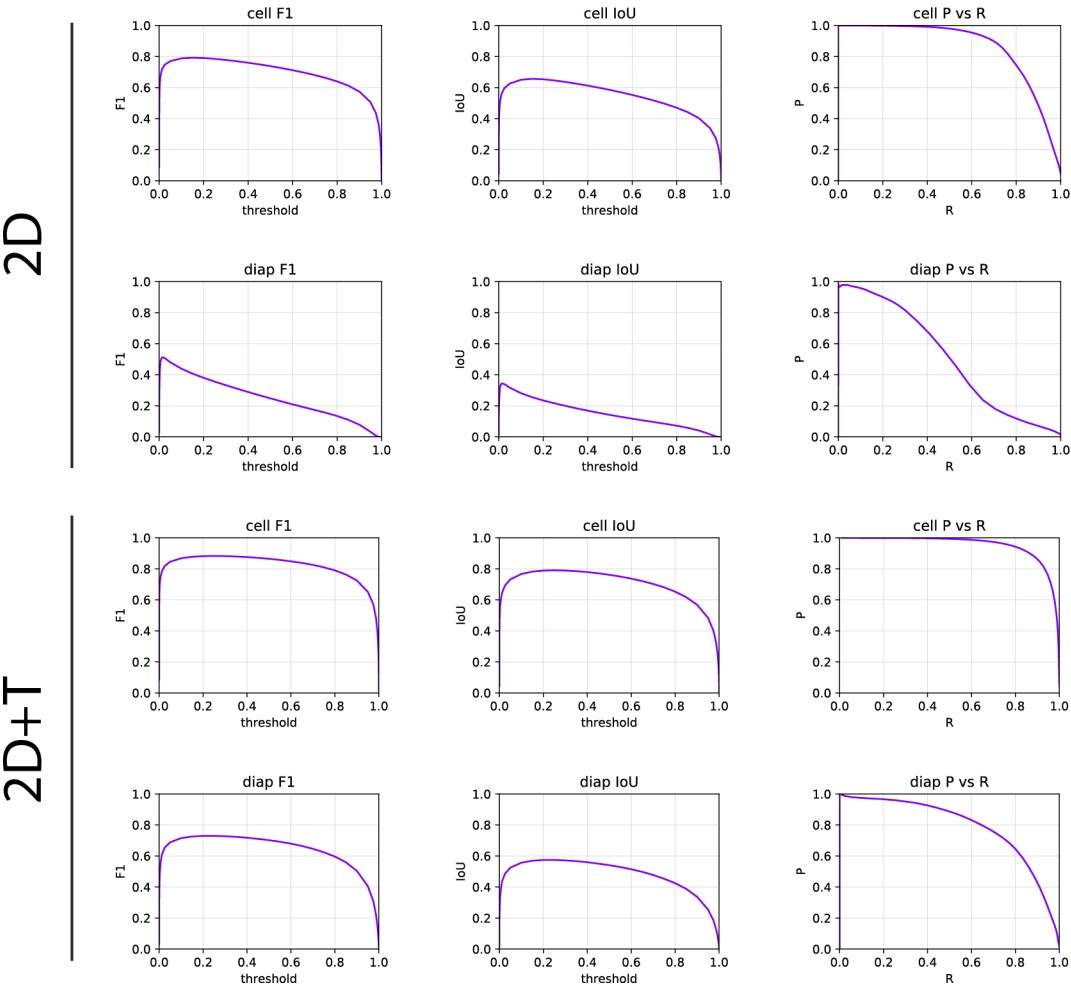

**Appendix 1—figure 2.** Comparison of the 2D and 2D+T-cell segmentation models' performance for the T-cell mask and transmigration mask prediction. F1, Jaccard index and AP metrics are shown.

**Appendix 1—table 1.** Architecture of the 2D fully convolutional model for T-cell segmentation.

| # | Operation | Kernel | Stride | DO rate | Output size | Diagram |
|---|---|---|---|---|---|---|
| 1 | Input | | | | 256 × 256 × 1 | |
| 2 | Conv+LReLU | 7 × 7 | | | 256 × 256 × 64 | |
| 3 | Conv+BN+DO+LReLU | 3 × 3 | | 0.2 | 256 × 256 ×64 | |
| 4 | MP | | 2 × 2 | | 128 × 128 × 64 | |
| 5 | Conv+BN+DO +LReLU | 3 × 3 | | 0.2 | 128 × 128 ×128 | |
| 6 | Conv+BN+DO+LReLU | 3 × 3 | | 0.2 | 128 × 128 ×128 | |
| 7 | MP | | 2 × 2 | | 64 × 64 × 128 | |
| 8 | Conv+BN+DO +LReLU | 3 × 3 | | 0.2 | 64 × 64 × 256 | |
| 9 | Conv+BN+DO +LReLU | 3 × 3 | | 0.2 | 64 × 64 × 256 | |
| 10 | MP | | 2 × 2 | | 32 × 32 × 256 | |
| 11 | Conv+BN+DO+LReLU | 3 × 3 | | 0.2 | 32 × 32 × 512 | |
| 12 | TConv+BN+LReLU | 3 × 3 | 2 × 2 | | 64 × 64 × 256 | |
| 13 | Concat(12, 9) | | | | 64 × 64 × 512 | |
| 14 | Conv+BN+DO +LReLU | 3 × 3 | | 0.2 | 64 × 64 × 256 | |
| 15 | Conv+BN+DO +LReLU | 3 × 3 | | 0.2 | 64 × 64 × 256 | |
| 16 | TConv+BN+LReLU | 3 × 3 | 2 × 2 | | 128 × 128 ×128 | |
| 17 | Concat(16, 6) | | | | 128 × 128 ×256 | |
| 18 | Conv+BN+ DO +LReLU | 3 × 3 | | 0.2 | 128 × 128 ×128 | |
| 19 | Conv+BN+DO +LReLU | 3 × 3 | | 0.2 | 128 × 128 ×128 | |
| 20 | TConv+BN+LReLU | 3 × 3 | 2×2 | | 256×256 × 64 | |
| 21 | Concat(20, 3) | | | | 256 × 256 ×128 | |
| 22 | Conv+BN+DO +LReLU | 3 × 3 | | 0.2 | 256 × 256 × 64 | |
| 23 | Conv+BN+DO +LReLU | 3 × 3 | | 0.2 | 256 × 256 ×64 | |
| 24 | ∟ Conv +Sigmoid | 1 × 1 | | | 256 × 256 × 2 | |
| 25 | Conv+BN + DO+LReLU | 3 × 3 | | 0.1 | 256 × 256 ×16 | |
| 26 | ∟ Conv +SSigmoid | 3 × 3 | | | 256 × 256 × 1 | |
| 27 | Concat(24, 26) | | | | 256 × 256 × 3 | |

Cell mask
Transmigration mask
Centroids

DO – dropout, Conv – Convolution, TConv – transposed convolution, MP – max pooling, BN – batch normalization, Concat – concatenation of outputs of the specified layers, LReLU – leaky rectified linear unit with $\alpha$=0.05. For the centroids prediction, we scale the output to prevent saturation: SSigmoid(x)=1.1 * Sigmoid(x) – 0.5.

**Appendix 1—table 2.** Architecture of the 2D+T fully convolutional model for T-cell segmentation.

| # | Operation | Kernel | Stride | DO rate | Output size | Diagram |
|---|-----------|--------|--------|---------|-------------|---------|
| 1 | Input | | | | 5 × 256 × 256 × 1 | |
| 2 | Conv+LReLU | 1 × 7 × 7 | | | 5 × 256 × 256 × 64 | |
| 3 | Conv+BN+DO+LReLU | 3 × 3 × 3 | | 0.2 | 5 × 256 × 256 ×128 | |
| 4 | └ ZCrop | | | | 1 × 256 ×256 ×128 | |
| 5 | ZCrop | | | | 3 × 256 ×256 ×128 | |
| 6 | └ MP | | 1 × 2 × 2 | | 3 × 128 × 128 × 128 | |
| 7 | Conv+BN+DO +LReLU | 1 × 3 × 3 | | 0.2 | 3 × 128 × 128 ×128 | |
| 8 | Conv+BN+DO+LReLU | 1 × 3 × 3 | | 0.2 | 3 × 128 × 128 ×128 | |
| 9 | └ ZCrop | | | | 1 ×128 ×128 ×128 | |
| 10 | └ MP | | 1×2 × 2 | | 3 × 64 × 64 × 128 | |
| 11 | Conv+BN+DO +LReLU | 1 × 3 × 3 | | 0.2 | 3 × 64 × 64 × 256 | |
| 12 | Conv+BN+DO+LReLU | 3 × 3 × 3 | | 0.2 | 3 × 64 × 64 × 512 | |
| 13 | ZCrop | | | | 1 × 64 × 64 × 512 | |
| 14 | MP | | 1 × 2× 2 | | 1 × 32 × 32 × 512 | |
| 15 | Conv+BN+DO +LReLU | 1 × 3 × 3 | | 0.2 | 1 × 32 × 32 × 256 | |
| 16 | Conv+BN+DO+LReLU | 1 × 3 × 3 | | 0.2 | 1 × 32 × 32 × 256 | |
| 17 | MP | | 1 × 2 × 2 | | 1 × 16 ×16 × 256 | |
| 18 | Conv+BN+DO+LReLU | 1 × 3 × 3 | | 0.2 | 1 × 16 × 16 × 512 | |
| 19 | TConv +BN + LReLU | 1 × 3 × 3 | 1 × 2 × 2 | | 1 × 32 × 32 × 256 | |
| 20 | Concat(19, 16) | | | | 1 × 32 × 32 × 512 | |
| 21 | Conv +BN + DO +LReLU | 1 × 3 × 3 | | 0.2 | 1 × 32 × 32 × 256 | |
| 22 | TConv +BN + LReLU | 1 × 3 × 3 | 1 × 2 × 2 | | 1 × 64 × 64 × 128 | |
| 23 | Concat(22, 13) | | | | 1 × 64 × 64 × 256 | |
| 24 | Conv +BN + DO +LReLU | 1 × 3 × 3 | | 0.2 | 1 × 64 × 64 ×128 | |
| 25 | Conv +BN + DO +LReLU | 1 × 3 × 3 | | 0.2 | 1 × 64 × 64 ×128 | |
| 26 | TConv +BN + LReLU | 1 × 3 × 3 | 1 × 2 × 2 | | 1 × 128 ×128×128 | |
| 27 | Concat(26, 9) | | | | 1 × 128 × 128 × 256 | |
| 28 | Conv +BN + DO +LReLU | 1 × 3 × 3 | | 0.2 | 1 × 128 × 128 × 128 | |
| 29 | Conv +BN + DO +LReLU | 1 × 3 × 3 | | 0.2 | 1 × 128 × 128 × 128 | |
| 30 | TConv +BN + LReLU | 1 × 3 × 3 | 1 × 2 × 2 | | 1 × 256 × 256 × 64 | |
| 31 | Concat (30, 4) | | | | 1 × 256 × 256 × 192 | |
| 32 | Conv +BN + DO +LReLU | 1 × 3 × 3 | | 0.2 | 1 × 256 × 256 × 64 | |
| 33 | Conv +BN + DO +LReLU | 1 × 3 × 3 | | 0.2 | 1×256 × 256 × 64 | |
| 34 | └ Conv +Sigmoid | 1 × 1 × 1 | | | 1 × 256 × 256 × 2 | |
| 35 | Conv +BN + DO +LReLU | 1 × 3 × 3 | | 0.1 | 1 × 256 × 256 × 16 | |
| 36 | └ Conv +SSigmoid | 1 × 3 × 3 | | | 1 × 256 × 256 × 1 | |
| 37 | Concat (34, 36) | | | | 1 × 256 × 256 × 3 | |

DO – dropout, Conv – Convolution, TConv – transposed convolution, MP – max pooling, BN – batch normalization, Concat – concatenation of outputs of the specified layers, ZCrop – crop center along depth dimension, LReLU – leaky rectified linear unit with $\alpha = 0.05$. For the centroids prediction, we scale the output to prevent saturation: SSigmoid(x)=1.1 * Sigmoid(x) – 0.5.

## Image data processing

For T-cell segmentation we followed a procedure similar to the one used for training data preprocessing. We first used the 2D model to localize T cells in individual image frames. We then dilated the predicted T-cell masks by 7 pixels, such that the bright halo on the phase-contrast images surrounding the T cells arresting and migrating on top of the pMBMEC monolayer was enclosed in the masked region.

Next, we performed image histogram normalization of the image frames based on the brightness distribution of the pMBMECs. For this we used the dilated T-cell masks obtained in the previous step. The same histogram reference dataset previously used for the model training was applied. Histogram normalization needed to be performed separately for the image frames obtained during the accumulation phase and the physiological flow phase of the assay. This is due to the overall image brightness increase after increasing the flow rate due to slight change of the sample position.

The histogram-normalized image sequence was then aligned based on the pMBMECs, using the mask imaged with the same dilated T-cell mask. Frame alignment (*Vladymyrov et al., 2020*) was performed by 5 crops the size of 512×512 pixels each in the corners and in the middle of the frame. We first measured offsets $dr_{c,i \to j}$ and offset errors $\sigma_{c,i \to j}$ between pairs of image frame crops $c$ at timepoints $i$ and $j, \forall i,j = i...i + 10$. Then, we employed global solving to find the absolute position $r_{c,i}$ and the respective position errors $\sigma_{c,i}$ for each crop $c$. Finally, we obtained the absolute positions of the entire frames by weighting the positions obtained by the crops:

$$ r_i = \frac{\sum_c \frac{r_{c,i}}{\sigma_{c,i}}}{\sum_c \frac{1}{\sigma_{c,i}}}. $$

The sequence of histogram-normalized and aligned image frames was then fed into the trained 2D+T segmentation model to obtain the T cell and transmigration probability maps, as well as the T-cell centroids. To this end we processed the image sequences in patches of 512×512×5 pixels, with an overlap of 183 pixels. The size of the receptive field of our T-cell segmentation model governed this overlap. Notably, while the models were trained on image crops of 256×256 pixels, the fully convolutional architecture allows the inference to be performed on any input size. Input size is thus limited solely by the available GPU memory.

The separate tiles of the tiled image sequence acquisition were aligned by overlapping parts of the raw image data. Predicted maps were then stitched together, where maximum predicted probability values were taken for overlapping tile regions.

To perform T-cell segmentation, we obtained T-cell masks from the T-cell probability map by applying a threshold of 0.33. Seed points were obtained from the T-cell centroid map with a threshold of 0.16. These threshold values maximized the F1 score of the T-cell mask prediction, as seen in *Appendix 1—figure 2*. The segmentation was then performed with a watershed algorithm based on the T-cell mask and the seed points. A stitched phase-contrast image sequence can be seen in *Video 1* and overlayed with the segmented cells and highlighted transmigration mask in *Video 2*.

## Appendix 2

### T-cell tracking

#### Linking T cells

As the first step, we identified all possible connections of T cells between the time frames. We constructed a graph representation of the datasets, such that graph nodes corresponded to the T cells at all time frames, and the edges in the graph corresponded to links connecting the nodes adjacent in space and time. Along the tracking procedure, we then sought to identify links connecting the same T cell over time. In cases where T cells touched each other, i.e., when several centroids were detected within the same T-cell mask or adjacent T-cell masks with distances below 10 μm (i.e. potentially over-segmented cells), we considered them as one node. At this step, we accounted for this under-segmentation by introducing the node multiplicity number $m$ equal to the number of constituting components. These adjacent cells were assigned to separate tracks after the reliable track segments of isolated T cells were constructed (see Resolving global track consistency section below).

The links between T cells were searched within a radius $R = dt \cdot 15\,\mu m + 10\,\mu m$, where time difference $dt$ is between 1 and 3 time frames (*Figure 3C*).

#### Track segment search

This step aimed to find continuous track segments of T cells or under-segmented groups of T cells crawling on top of or below the pMBMEC monolayer without accelerated movement segments on the T-cell track characterized by rapid T-cell displacements. The whole dataset of T cells across all timepoints was represented as a graph. Each vertex corresponds either to a T cell (multiplicity $m$=1) or a group of potentially under-segmented T cells ($m$>1). The vertices are connected according to links obtained at the linking step. Track segments were found by performing global optimization to find consistent connectivity of vertices across the timepoints by employing an approach inspired by the conservation tracking algorithm summarized in *Schiegg et al., 2013*. Each vertex was represented as a set of incoming (Left, L) and outgoing (Right, R) links, and additional 'not-connected' (NC) left and right links (*Figure 3C and D*, *Appendix 2—figure 1A*, *Appendix 2—table 1*). The optimization procedure selected the links that were most likely to be connections between the same T cell in different time frames. The total number of selected links for each vertex on the left and right sides was limited to vertex multiplicity m. Optimization was performed using the CP-SAT constrained optimization procedure using the open-source OR-Tools library (*Perron and Furnon, 2021*).

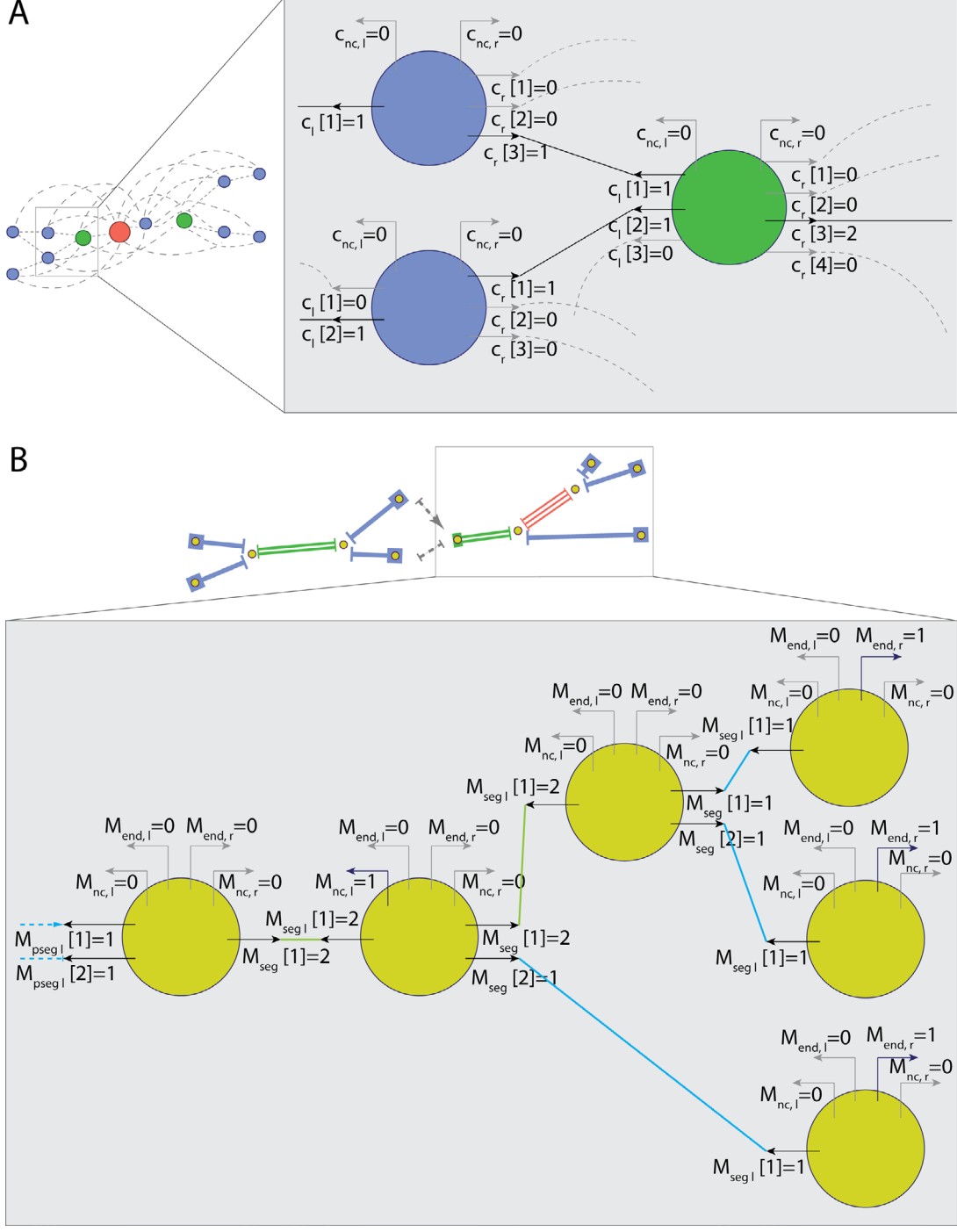

**Appendix 2—figure 1.** Parametrization of nodes connections for tracking. (**A**) Representation of nodes for track segment search. Connection multiplicity for each link is obtained with global optimization. (**B**) Representation of vertices and segments for global multiplicity consistency optimization. Connection multiplicity for each segment attached to a vertex is obtained with global optimization.

**Appendix 2—table 1.** Node variables used in global optimization during link search.

| Name | Description | Type | Range |
|---|---|---|---|
| $c_{l/r}[i]$ | Multiplicity of the [i]th node connection on the left/right side | Int | 0–m |

*Appendix 2—table 1 Continued on next page*

*Appendix 2—table 1 Continued*

| Name | Description | Type | Range |
|------|-------------|------|-------|
| $c_{w_l/r}[i]$ | $i$th connection on the left/right is selected | Int | 0–1 |
| $c_{nc_l/r}$ | Not-connected (NC) connection on the left/right is selected | Int | 0–1 |
| $d_{\|lr}$ | Absolute difference between right and left multiplicity sum | Int | 0–m |

We employed the following constraints for each node.
Constrain 'not-connected' variable:

$$c_{nc\_l} = \begin{cases} 0 & \text{if } \sum_i c_l[i] > 0 \\ 1 & \text{otherwise} \end{cases}$$

Limit total 'multiplicity' of node connections:

$$1 \leq c_{nc\_l} + \sum_i c_l[i] \leq m$$

Link the 'selected' flag of a connection with its multiplicity:

$$c_{w\_l}[i] = \begin{cases} 1 & \text{if } c_l[i] > 0 \\ 0 & \text{otherwise} \end{cases}$$

And similarly for the right side.
If the node has connection candidates on both sides, then evaluate the multiplicity difference:

$$d_{abs\_lr} = \begin{cases} 0 & \text{if } \sum_i c_r[i] = 0 \vee \sum_i c_l[i] = 0 \\ \left| \sum_i c_r[i] - \sum_i c_l[i] \right| & \text{otherwise} \end{cases}$$

On each edge between connected nodes, the multiplicity on both sides should be the same:

$$c_{r, \text{left node}}\left[i_{\text{connection to right node}}\right] = c_{l, \text{right node}}\left[j_{\text{connection to left node}}\right]$$

Then, the loss function for each node is

$$L_n = d_{abs\_lr} \cdot w_m + \sum_{l,r} c_{nc} \cdot w_{nc} + \sum_i c_{w\_r}[i] \cdot w[i]$$

This loss has three terms. The first one penalizes differences between connection multiplicity on the left and right sides. The second penalizes solutions where the node is not connected on either side. The last term weights each selected connection to a node on the right. Here, $w[i]$ is the weight of the connection to the corresponding right node. Importantly, connection weighting is independent of the connection multiplicity. This design choice was made explicitly since node multiplicity estimates available at this stage are preliminary.

To account for T-cell detection inefficiency at the edge of the FoV, the penalty for not-connected nodes $w_{nc}$ is attenuated close to the FoV edge. To this end the weight of a not-connected node exponentially decays when the T cell is closer than $d_0 = 20\,\mu m$ to the edge:

$$w_{nc} = f_{edge} \cdot w_{nc\_0},$$

where

$$f_{edge} = \begin{cases} 1 & if\ d > d_0 \\ exp\left(\frac{d_0 - d}{d_c}\right) & otherwise \end{cases}$$

$d_c$ is chosen such that $w_{nc} = w_{nc\_0}$ if $d = d_0$ and $w_{nc} = 1$ if $d = r_{cell}$, where the *T* cell radius (half of the crawling T-cell length) $r_{cell} = 10\,\mu m$.

The constants $w_m = 3$, $w_{nc} = 9$ were chosen empirically to achieve reliable reconstruction of long tracks.

Connections between nodes $c_1$ and $c_2$ at time frames $t_1$ and $t_2$ were correspondingly considered if they are at most 3 time frames apart $1 \leq t_2 - t_1 \leq dt_{max} = 3$, according to the output of the linking step. The connection weight $w$ between these nodes was evaluated as the negative log-likelihood (NLL) of the connection between these nodes:

$$w = w_v + w_A + w_{dt}$$

$w_v = \frac{1}{2}\left(\frac{v - v_\mu}{\sigma_v}\right)^2$ is the NLL of the crawling speed (path length over time), where $v = \frac{r_{12}}{(t_2 - t_1) \cdot \Delta t}$ is the speed estimate based on the positions $r_1, r_2$ of the closest of the T cells constituting nodes $c_1, c_2$. $\Delta t = 10s$ is the time step between consecutive time frames. $v_{\mu m} = 9\,\mu m/min$ and $\sigma_v = 8.5\,\mu m/min$ are prior population mean and standard deviation of T-cell crawling speed.

$w_a = \frac{1}{2}\left(\frac{\Delta A}{\sigma_A}\right)^2$, where $\Delta A = \min_{cells\ in\ c_1, c_2} 2\frac{A_1 - A_2}{A_1 + A_2}$, $A_1$ and $A_2$ are the areas of the detected constituting cells of nodes $c_1, c_2$. $\sigma_A = 0.397$ is the prior standard deviation of $\Delta A$ estimated from the data.

$w_{dt} = w_{missing} \cdot (t_2 - t_1 - 1)$ penalizes connections between nonconsecutive time frames while accounting for possible T-cell detection inefficiency. The constant $w_{missing} = 2.3$ was chosen empirically.

To reduce the time required for this optimization, the procedure was performed independently for each connected by link components of the dataset graph.

## Resolving global track consistency

In this step, the scope of T-cell tracking is shifted from individual nodes (representing T cells at particular time frames and groups of under-segmented T cells) to the track segments – unambiguous sequences of nodes and vertices at the endpoints of the segments. These are track start and end points, points of merging and separation of track segments in case of under-segmentation, as well as ambiguous points on a track. The latter were identified by sudden T-cell displacement, a hallmark of detaching and reattaching T cells and T cells transitioning from properly segmented to under-segmented T cells or vice versa (*Figure 3D and E*).

First, the segments were searched as a sequence of links between nodes connected to only one node on the following time frame or nodes branching to multiple nodes and later rejoining to one node. This corresponded to over-segmentation at some points of the track or an under-segmented group of T cells shortly separating to being properly segmented before rejoining. These branching and rejoining node sequences were considered one segment at this stage. The vertices were the starting and end nodes of the segments, including the branching points between the segments. This significantly simplified the graph representation of the T-cell tracks. Next, the segments were split according to additional criteria, such as large displacements caused by T-cell detachment or intersection with another T cell. Empirically we have chosen the displacement threshold for splitting a segment to be $d > d_{thres} = \mu_d + 0.5\,\sigma_d$, where $\mu_d$, $\sigma_d$ are, respectively, the mean and standard deviation of T-cell displacements between consecutive time frames along the given segment. The node with the larger area was assigned as the new vertex. With each segment we associated the segment multiplicity $M$. Intuitively, this corresponds to the conservation of T-cell number along a track segment. At first, it was estimated as $M_0 = \underset{t}{Median}\left(\Sigma_{cells}m\right)$ median multiplicity of constituting nodes along the track (*Figure 3D*).

Next, we searched for potential missing track segments due to accelerated T-cell movement under flow and missing links in the track crossing points. Both are characterized by considerable displacement length such that they were not detected during the linking step. Thus, we will refer to both as 'jumps' in this section. The two categories differ in the distribution of displacement length and direction. To perform this search, the likelihood $w_{ps}$ of the potential segments between vertices according to the average T-cell 'jump' length and direction for displacement caused by flow and

T-cell mask fusion due to under-segmentation was evaluated. In the case of jumps caused by under-segmentation, the square root of the displacement is approximately normally distributed. Thus, the NLL of a potential segment was estimated as

$$w_{sj0} = c_{sj} \frac{1}{2} \left( \frac{\sqrt{d} - \mu_{sj}}{\sigma_{sj}} \right)^2 ,$$

where $\mu_{sj}, \sigma_{sj}$ are, respectively, the mean and standard deviation of the distributions of the square root of the displacement $\sqrt{d}$ and

$$c_{sj} = \begin{cases} 1 & if \ \sqrt{d} < \mu_{sj} \\ 2 & otherwise \end{cases}$$

(see *Appendix 2—figures 2A*). The time duration of this potential segmentation-caused jump segment is also penalized, leading to the total NLL:

$$w_{sj} = \frac{1}{\sqrt{2}} \left( w_{sj0} + c_{sj,t} \left( dt - 1 \right) \right) ,$$

where $dt$ is segment duration in time frames and $c_{sj,t} = 1$.

For the T-cell jumps caused by flow, we empirically estimated (see *Appendix 2—figure 2B–D*) the shape of the displacement likelihood and parametrized it in the following way:

$$w_{fj0} = \begin{cases} \frac{1}{2} \left( \frac{d/dt - \mu_{fj}}{\sigma_{fj}} \right)^2 + c_{fj}\sqrt{dt} \dfrac{\arctan\left( \left| \frac{dy}{dx} \right| \right)}{\pi} & if \ dx < 0 \ and \ \left| dy \right| < \left| dx \right| \\ 100 & otherwise \end{cases}$$

where $d$ is the displacement in time $dt$, $\mu_{fj} = 25\,\mu m, \sigma_{fj} = 12.9\,\mu m$, and $c_{fj} = 36$.

$$w_{fj} = \frac{1}{\sqrt{2}} \left( w_{fj0} + c_{fj,t} \left( dt - 1 \right) \right) ,$$

where $dt$ is segment duration in time frames and $c_{sj,t} = 3$.

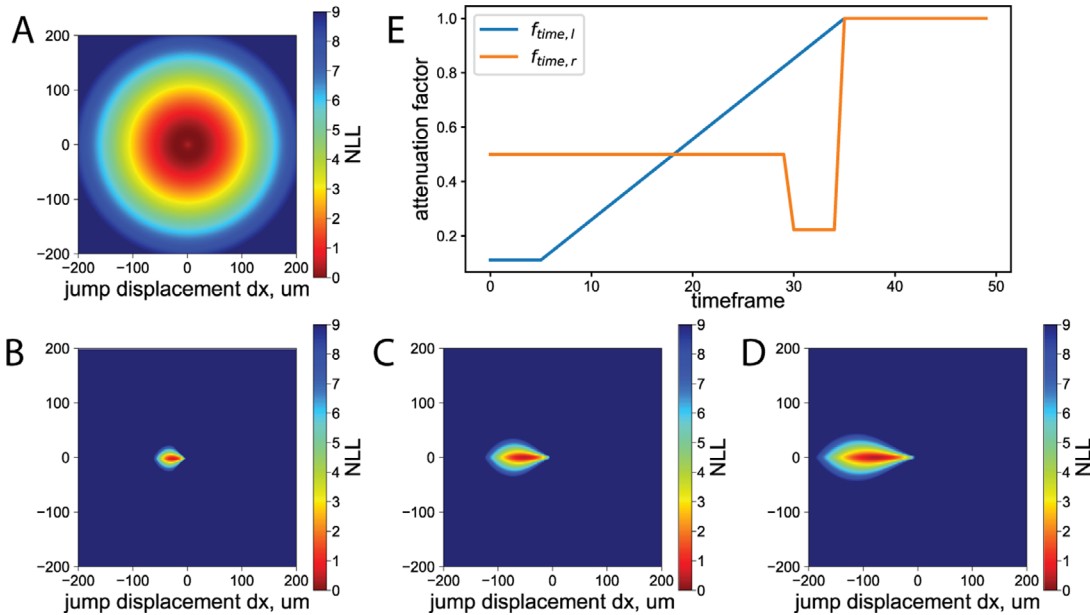

**Appendix 2—figure 2.** Under-flow tracking parametrization. (**A**) Negative log-likelihood $w_{sj0}$ of the potential T-cell jump segments due to under-segmentation. (**B–D**) Negative log-likelihood $w_{fj0}$ of the potential T-cell
*Appendix 2—figure 2 continued on next page*

*Appendix 2—figure 2 continued*
jump segments due to the flow. (**B**) $dt$=1, (**C**) $dt$=2, (**D**) $dt$=3. (**E**) Attenuation of the vertex 'not-connected' weight with time allows accounting for the T-cell accumulation phase at time frames 5–30 and increase of the flow to physiological level at time frame 30. The blue curve shows an attenuation factor on the left side, i.e., corresponding to track start, and the orange curve shows an attenuation factor on the right side corresponding to the end of the track due to cell detachment under the flow.

Since potential segments only with NLL below the not-connected NLL $w_{nc}$ can be selected by the optimization procedure, the segments with NLL above $w_{nc}$ were discarded. The remaining potential segments of the types 'segmentation jump' and 'flow jump' were appended to the graph (*Figure 3E*, *Appendix 2—figure 1B*).

Next, we found the optimal multiplicity on each segment $M_{seg}$ and potential segment $M_{pseg}$. We identified T-cell jumps along the track as the potential segments where the multiplicity $M_{pseg}$ was found to be nonzero. To this end we performed optimization under the constraint of global multiplicity consistency analogous to the one described above for the track segment search. Optimization was performed on each subgraph connected by segments independently. Here, we demanded the same multiplicity on both ends of the segment $M_{seg,l} = M_{seg,r}$, consistent with corresponding multiplicity on the connected vertex $M_{seg[i],r} = M[i]_{vtx,l}$, and same multiplicity on both sides of the vertex,

$$M_{end\_l} + M_{nc\_l} + \sum_{i,l} M_{seg}[i] + \sum_{i,l} M_{pseg}[i] = M_{end\_r} + M_{nc\_r} + \sum_{i,r} M_{seg}[i] + \sum_{i,r} M_{pseg}[i],$$

where $M_{end}$ is the multiplicities in case the vertex is a track endpoint, $M_{nc}$ is the resolved missing multiplicity at the vertex, and $M_{seg}[i], M_{pseg}[i]$ are the multiplicities of the segments and potential segments correspondingly that are attached to the vertex on left and right sides. Left and right endpoint multiplicity $M_{end_{l/r}} = 0$ if any segment is connected on the corresponding side. We constrained $0 \leq M_{nc} \leq M_{max}$, $f_0 \leq M_{seg}[i] \leq M_0 + M_{max}$, and $0 \leq M_{pseg}[i] \leq M_{max}$ where $M_{max} = 5$. We did not consider intersections of more than 5 T cells, as those could not be reliably identified on the opposite ends of the intersections by their motility parameters (see the Intersection resolving section), and in our experiments we found this value to be sufficient.

Optimization was subjected to minimizing the loss function in each node:

$$L_n = \sum_{l,r} M_{end} \cdot w_{end} + \sum_{l,r} M_{nc} \cdot w_{nc} + \sum_i \left( M[i] - M_0[i] \right) \cdot w_{over} + \sum_i c_{ps}[i] \cdot w_{ps}[i]$$

Here, we distinguished missing multiplicity on a vertex $M_{nc} \neq 0$, corresponding to T cells detaching in a group of under-segmented T cells when some of the T-cell tracks continued further from the endpoint of the track, where $M_{end} \neq 0$. The first term of the loss function penalized higher multiplicity at the endpoints of the track, encouraging solutions with T-cell tracks being properly segmented in the endpoints. Here, $M_{end}$ is the multiplicity at the corresponding endpoint, and the $w_{end} = 6$.

Putting additional weight on the missing multiplicity on a vertex in the middle of the graph but not on the first node of a track as opposed to all nodes at non-first time frames of the dataset allowed us to account for flow causing T-cell tracks to start and end at any time frame due to T-cell attachment during the accumulation phase and detachment of T cells carried away by the flow. Thus, the second term penalizes the missing multiplicity on either side of the vertex with the coefficient $w_{nc}$: $w_{nc} = f_{edge} f_{time} w_{nc\_0}$,

where $w_{nc\_0} = 9$, $f_{edge}$ corresponds to the attenuation close to the edges of the FoV as described above. We considered the T-cell accumulation phase under low flow by attenuating the weight of track start and end $w_{nc}$ with time (*Appendix 2—figure 2*):

$$f_{time,l} = \begin{cases} \dfrac{1}{9} & \text{if } t < t_{acc.start} \\ \dfrac{1}{9} + \dfrac{8}{9} \dfrac{t - t_{acc.start}}{t_{acc.end} + 5 - t_{acc.start}} & \text{if } t_{acc.start} \leq t < t_{acc.end} + 5 \\ 1 & \text{otherwise} \end{cases}$$

$$f_{time,r} = \begin{cases} \frac{1}{2} & \text{if } t < t_{acc.end} \\ \frac{2}{9} & \text{if } t_{acc.end} \leq t < t_{acc.end} + 5 \\ 1 & \text{otherwise} \end{cases}$$

This corresponded to a higher probability for track to start or end (T cell attaching/detaching from the pMBMECs monolayer) during the accumulation phase, as well as a high probability for the track to end during the 5 frames after the flow was increased to the physiological level.

The third term of the loss function encouraged minimal adjustments to the multiplicity estimates by penalizing the difference between the initial multiplicity estimate on T-cell track segments and the final solution with $w_{over} = 1$.

The last term accounted for the potential segments. $c_{ps}[i] = 1$, if the multiplicity $M_{pseg}$ on the corresponding potential segment was found to be above zero, and $c_{ps}[i] = 0$ when the multiplicity was zero. $w_{ps}[i]$ is the NLL of the potential segment, i.e., either $w_{fj}$ or $w_{sj}$ for the flow- or segmentation-caused jump segments correspondingly. All the coefficients were chosen empirically to optimize the detection of T-cell jumps.

Finally, we eliminated short and thus unreliable segments, as well as segments for which the multiplicity was found to be $M = 0$ (**Figure 3F**). To this end, we iteratively removed vertices, which were not connected on one side and connected to a branching vertex of the graph on the other side by a segment with less than 6 nodes, as well as the connecting segment, and repeated the global multiplicity optimization until reaching convergence. To reduce the wall time required for the optimization, it was again performed independently for each connected component of the dataset graph.

Afterward, we separated the segments into two categories, namely segments with multiplicity $M = 1$, i.e., tracks of isolated T cells, and segments with multiplicity $M > 1$, i.e., tracks of under-segmented groups of T cells, where tracks of several T cells intersected (**Figure 3G**).

In the end, we extended the tracks of isolated T cells into the intersection if the branching vertex had a multiplicity of $n$ and was splitting in exactly $n$ segments with multiplicity $M = 1$ (**Figure 3G**). Specifically, we performed the segment extension if there was an unambiguous assignment of cells constituting the branching vertex to the segments connected to the vertex, i.e., that the weight $w_{i,j}$ of assigning $i$th cell belonging to the vertex to $j$-th segment connected to the vertex satisfies simultaneously $\underset{j}{argmin}\, w_{i_a,j} = j_a$ and $\underset{i}{argmin}\, w_{i,j_a} = i_a$. To this end, we evaluated the standard deviation $\sigma_j$ of the distance from the T cell along the $j$-th segment to the line obtained by linear extrapolation of the cell coordinated at the nearest 6 time frames. We then obtained the expected T-cell position $r_{j,t}$ by linear extrapolation of the $j$-th segment to the time frame $t$ of the vertex using the nearest 6 time frames of the segment. The assignment weight was evaluated as the NLL:

$$w_{i,j} = \frac{1}{2}\left(\frac{dr_{i,j}}{\sigma_j}\right)^2,$$

where $dr_{i,j} = r_{i,t} - \widehat{r_{j,t}}$ is the distance between position $r_{i,t}$ of the $i$th cell belonging to the vertex at time frame $t$ and the expected cell position $\widehat{r_{j,t}}$.

## Intersection resolving

The last step required to obtain reliable T-cell tracks is resolving track intersections, i.e., identifying track segments corresponding to the same T cell before and after the under-segmented track region (**Figure 3H**). Even though T-cell motility parameters were found to be quite similar within the population investigated, evaluating them on longer segments allowed us to identify the correspondence between segments of the same T-cell track.

To this end, we evaluated the NLL of potential assignments of track segments according to the distance between their closest points and mean T-cell crawling speed (cell track path length over time), migration speed (cell displacement over time), directionality, cell ellipticity, and cell area. We estimated these parameters employing the Bayesian estimator with prior expected values evaluated on tracks of isolated T cells obtained after resolving the global track consistency. We then used F-statistic to assess the connection probability and to obtain the corresponding NLL.

We then employed the linear assignment problem (LAP) approach to identify the optimal segment assignments across the track intersections. To ensure that reliable connections were made, instead of relying on the assignment with the first solution for all tracks, we instead connected only the segments with the $NLL < NLL_{max,i}$ progressively increasing $NLL_{max,i}$ in $\{1, 3, 6, 9, \infty\}$ and repeating the parameter estimation for the remaining unresolved track segments and the LAP solving. This ensured that the T-cell motility parameters used for segment matching were estimated on the longer, already connected T-cell track segments.

This workflow allowed us to determine fully resolved T-cell tracks, with either assigned T-cell position or identified locations of rapid T-cell displacement, cell under-segmentation time span, and missing time frames due to potential T-cell detection inefficiency (*Figure 3H*).

## T-cell migration analysis

Next, we performed the T-cell migration analysis based on the reconstructed T-cell tracks. We selected tracks inside the fiducial area of the FoV, namely coordinates of the T cell at all timepoints along the track were located at least 25 μm away from the bounding box enclosing all segmented T cells. Next, tracks of T cells touching another T cell at the end of the assay acquisition were excluded since T cells directly adjacent to each other can hide the start of T-cell transmigration across the pMBMEC monolayer and thus compromise correct detection and quantification of this step. Additionally, only tracks with T cells assigned in at least 6 time frames during the physiological flow phase. We also require T cells to be assigned for at least 75% of time frames along the track. Under-segmented parts of T-cell tracks were not considered. Examples of selected tracks can be seen in *Video 3*.

First, we counted the number of neighboring tracks, i.e., T cells approaching another T cell closer than 24 μm (1.2 of a crawling T-cell length).

T-cell transmigration across the pMBMEC monolayer was detected based on the inferred T-cell transmigration coefficient $t_c$. To reduce the noise, we obtained filtered time series of transmigration coefficients $t_{c,f}$ by applying a Gaussian filter with $\sigma = 2.5$ time frames to the $t_c$ time series. We defined the T cell to the preliminary categories 'full transmigration' and category 'partial transmigration' with $t_{c,f} > 0.75$ and $t_{c,f} > 0.3$ respectively. The 'full transmigration' and 'partial transmigration' Boolean masks were constructed according to the evolution of these two categories along the track (*Figure 3—figure supplement 1*). To further reduce the noise, we also closed short gaps (sequence of false values) in the transmigration masks with length 2 and 3 time frames for the 'partial' and 'full' transmigration respectively. Then, we removed short sequences of true values in the masks of 4 and 3 time frames correspondingly.

Based on these preliminary masks, we detected the additional categories of uncompleted transmigration, direct transmigration, and reverse transmigration categories (*Figure 1D*). During uncompleted transmigration, the T cell starts the transmigration process and later on retracts the protrusions to continue crawling above the pMBMEC monolayer. Direct transmigration covers the period during which the T cell transmigrates below the pMBMEC monolayer. In contrast, during reverse transmigration, the T cell previously located below the pMBMEC monolayer reversely migrates back to the luminal side of the pMBMEC monolayer (*Figure 3—figure supplement 1*). The start, duration, and number of transmigration attempts were evaluated for each transmigration category.

We employed an additional classifier to discriminate a migrating T cell from cellular debris or particles (=not-a-T-cell) to reject the cellular debris misclassified as T cells during the segmentation step. This classification was performed based on the measured mean and median T cell areas along the T-cell tracks after the flow was increased to physiological levels. Specifically, we trained a logistic regression model using the scikit-learn package (*Pedregosa et al., 2011*). The model was trained on 105 manually annotated T cells by an experimenter and subsequently validated on 27 additional T cells. We have achieved validation accuracy of 96%. The tracks classified as 'not-a-T-cell' were discarded from the downstream analysis.

Due to occasional over-segmentation of the T cell during the transmigration process, a T cell could be misclassified as detaching instead of as transmigrated. To address this, we have trained a linear classifier to differentiate a detached T cell from a transmigrated T cell. In this case, the classification was performed based on the following parameters: mean $a_{mean}$

and median $a_{med}$ T-cell area along the part of the T-cell track during the physiological flow phase, $\frac{a_{last\,1}}{a_{mean}}$, $\frac{a_{last\,1}}{a_{med}}$, $\frac{a_{mean\,last\,5}}{a_{mean}}$, $\frac{a_{mean\,last\,5}}{a_{med}}$, where $a_{last\,1}$ and $a_{mean\,last\,5}$ were the T-cell area at the last and mean over the last 5 time frames of the T-cell track correspondingly, mean $p_{c,mean}$ and median $p_{c,med}$ T-cell detection probability as well as probabilities at the last time frame $p_{c,last\,1}$ and mean of last 5 time frames $p_{c,mean\,last\,5}$, and the mean transmigration coefficient over the last 5 time frames of the T-cell track $t_{c,mean\,last\,5}$.

If the track of a T cell remaining above the pMBMEC monolayer ended before the end of the assay and this T cell is classified as detaching, it was marked as a detached T cell accordingly. Following their transmigration across the pMBMEC monolayer, T cells continued to crawl below the pMBMEC monolayer in the microfluidic device but were no longer subjected to flow. If such tracks were ending before the end of the assay, they were labeled as 'tracking inefficiency' and were subsequently excluded from further analysis of the post-transmigration motility parameters.

Next, we created masks of tracking inefficiency and periods of accelerated movement, i.e., rapid displacement of the T cells on the pMBMEC monolayer caused by the flow. For each T-cell track during the physiological flow phase and prior to transmigration, we evaluated the median step speed $v_{step,med}$, i.e., the T-cell speed between sequential time frames and its median absolute deviation $MAD\,(v_{step})$. We evaluated the mean instantaneous speed $v_{inst,mean}$ and its standard deviation $\sigma_{v,inst}$, excluding steps where the speed was an outlier, i.e., $v_{inst} > v_{inst,med} + 3\,\sigma_{vinst\,med}$, where $\sigma_{v\,inst\,med} = 1.4826\,MAD\,(v_{inst})$. We labeled time span along the T-cell tracks as 'accelerated T-cell movement' if the instantaneous speed

$$v_{inst} > min\left(v_{inst,mean} + 3\,\sigma_{v,inst} \cdot v_{crawling\,max}\right)$$

and the instantaneous displacement $dr_{inst} > dr_{am}$, where $v_{crawling\,max} = 36\,\mu m/min$ and $dr_{am} = 8\,\mu m$. If a step had a time difference of more than 3 time frames, e.g., in the case of intersecting T-cell tracks, it was labeled as tracking inefficiency.

If the T-cell displacement was more than $20\,\mu m$ (length of a crawling T cell) during a cell track segment, irrespective of above or below the pMBMEC monolayer, the segment was marked as crawling. This is analogous to the criteria used during manual analysis (see the 'Materials and methods' section). The cell track was marked as probing if the T cell did not crawl before the first transmigration attempts.

Finally, for each track, we evaluated motility parameters for each of the following migration regimes: probing before the transmigration, crawling before the transmigration, all crawling above pMBMECs monolayer including T-cell crawling segments after the first transmigration attempts, all crawling below pMBMECs monolayer, whole T-cell track excluding accelerated movement and tracking inefficiency regions, as well as whole T-cell track. Specifically, we evaluated the following T-cell motility parameters: duration of each migration regime, the total vector and absolute displacements, the migration path length, the average migration speed (displacement over time), average crawling speed (path length over time), and finally the mean and standard deviation of the instantaneous speed. For the accelerated movement regime, we evaluated migration time, displacement, and average speed.

