## [Editor Report · eLife Assessment]

This work is **important** because it elucidates how immune cells migrate across the blood brain barrier. In the revised version of this study, the authors present a **convincing** framework to visualize, recognize and track the movement of different immune cells across primary human and mouse brain microvascular endothelial cells without the need for fluorescence-based imaging using microfluidic devices. This work will be of broad interest to the cancer biology, immunology and medical therapeutics fields.

---

## [Referee Report · Reviewer #1 (Public review)]

Summary:

It is evident that studying leukocyte extravasation in vitro is a challenge. One needs to include physiological flow, culture cells and isolate primary immune cells. Timing is of utmost importance and a reproducible setup is essential. Extra challenges are met when extravasation kinetics in different vascular beds is required, e.g., across the blood-brain barrier. In this study, the authors describe a reliable and reproducible method to analyze leukocyte TEM under physiological flow conditions, including this analysis. That the software can also detect reverse TEM is a plus.

Strengths:

It is quite a challenge to get this assay reproducible and stable, in particular as there is flow included. Also for the analysis, there is currently no clear software analysis program, and many labs have their own methods. This paper gives the opportunity to unify the data and results obtained with this assay under label-free conditions. This should eventually lead to more solid and reproducible results.

Also, the comparison between manual and software analysis is appreciated.

---

## [Referee Report · Reviewer #2 (Public review)]

Summary:

This paper develops an under-flow migration tracker to evaluate all the steps of the extravasation cascade of immune cells across the BBB. The algorithm is useful and has important applications.

Strengths:

The algorithm is almost as accurate as manual tracking and importantly saves time for researchers. The authors have discussed how their tool compares to other tracking methods.

Weaknesses:

Applicability can be questioned because the device used is 2D and physiological biology is in 3D. However, the authors have addressed this point in their manuscript.

---

## [Author Response]

The following is the authors’ response to the previous reviews.

**Public Reviews:**

**Reviewer #1 (Public review):**
Summary:It is evident that studying leukocyte extravasation in vitro is a challenge. One needs to include physiological flow, culture cells and isolate primary immune cells. Timing is of utmost importance and a reproducible setup essential. Extra challenges are met when extravasation kinetics in different vascular beds is required, e.g., across the blood-brain barrier. In this study, the authors describe a reliable and reproducible method to analyze leukocyte TEM under physiological flow conditions, including this analysis. That the software can also detect reverse TEM is a plus.Strengths:It is quite a challenge to get this assay reproducible and stable, in particular as there is flow included. Also for the analysis, there is currently no clear software analysis program, and many labs have their own methods. This paper gives the opportunity to unify the data and results obtained with this assay under label-free conditions. This should eventually lead to more solid and reproducible results.Also, the comparison between manual and software analysis is appreciated.Weaknesses:The authors stress that it can be done in BBB models, but I would argue that it is much more broadly applicable. This is not necessarily a weakness of the study but more an opportunity to strengthen the method. So I would encourage the authors to rewrite some parts and make it more broadly applicable.

We thank the Reviewer for this suggestion. The barrier properties of the BBB influence the dynamic behavior of T cells during their multi-step extravasation cascade. The crawling of CD4 T cells against the direction of blood-flow is e.g. a unique behavior of T cells on the BBB that is also observed in vivo(1-3). Nevertheless we fully agree that in principle UFMTrack is usable for studying in general immune cell interactions with endothelial monolayers under physiological flow. We have thus added a statement in the abstract and expanded the discussion to highlight availability of the framework and the potential necessary adaptations required when using UFMTrack for analyzing different experimental setups. Please also note, UFMTrack has been established as basic framework using the example of brain endothelial monolayers and one flow chamber devices while studying different immune cell subsets. The purpose of the publication is to make UFMTrack available to the community to address their specific questions.

(1) Kawakami, N., Bartholomäus, I., Pesic, M. & Kyratsous, N. I. Intravital Imaging of Autoreactive T Cells in Living Animals. *Methods Cell Biol.* 113, 149–168 (2013).

(2) Schläger, C., Litke, T., Flügel, A. & Odoardi, F. In Vivo Visualization of (Auto)Immune Processes in the Central Nervous System of Rodents. in 117–129 (Humana Press, New York, NY, 2014). doi:10.1007/7651_2014_150

(3) Haghayegh Jahromi, N. *et al.* Intercellular Adhesion Molecule-1 (ICAM-1) and ICAM-2 Differentially Contribute to Peripheral Activation and CNS Entry of Autoaggressive Th1 and Th17 Cells in Experimental Autoimmune Encephalomyelitis. *Front. Immunol.* 10, 3056 (2020).